Journal of Data-centric Machine Learning Research (2026) Submitted 08/23, 2025; Revised 12/18, 2025; Published 01/21, 2026

# Mobile-MMLU: A Mobile Intelligence Language Understanding Benchmark

**Sondos Mahmoud Bsharat**[‡*]               SONDOS.BSHARAT@MBZUAI.AC.AE

**Mukul Ranjan**[‡*]                          MUKUL.RANJAN@MBZUAI.AC.AE

**Aidar Myrzakhan**[‡*]                        AIDAR.MYRZAKHAN@MBZUAI.AC.AE

**Jiacheng Liu**[‡]                            JIACHENG.LIU@MBZUAI.AC.AE

**Bowei Guo**[‡]                               BOWEI.GUO@MBZUAI.AC.AE

**Shengkun Tang**[‡]                           SHENGKUN.TANG@MBZUAI.AC.AE

**Zhuang Liu**[♮]                              ZHUANGL@PRINCETON.EDU

**Yuanzhi Li**[♯]                              YUANZHIL@ANDREW.CMU.EDU

**Zhiqiang Shen**[‡†]                          ZHIQIANG.SHEN@MBZUAI.AC.AE

[‡] *VILA-Lab, Mohamed bin Zayed University of AI, Masdar City, Abu Dhabi, UAE*

[♮] *Princeton University, Princeton, NJ, USA*

[♯] *Meta, Menlo Park, CA, USA*

**Reviewed on OpenReview:** *https://openreview.net/forum?id=AaFRpxD6po*

**Editor:** Peter Mattson

## Abstract

Rapid advancements in large language models (LLMs) have increased interest in deploying them on mobile devices for on-device AI applications. Mobile users interact differently with LLMs compared to desktop users, creating unique expectations and data biases. Current benchmark datasets primarily target server and desktop environments, and there is a notable lack of extensive datasets specifically designed for mobile contexts. Additionally, mobile devices face strict limitations in storage and computing resources, constraining model size and capabilities, thus requiring optimized efficiency and prioritized knowledge. To address these challenges, we introduce **Mobile-MMLU**, a large-scale benchmark dataset tailored for mobile intelligence. It consists of 16,186 questions across 80 mobile-related fields, designed to evaluate LLM performance in realistic mobile scenarios. A challenging subset, **Mobile-MMLU-Pro**, provides advanced evaluation similar in size to MMLU-Pro but significantly more difficult than our standard full set. Both benchmarks use *multiple-choice*, *order-invariant* questions focused on practical mobile interactions, such as recipe suggestions, travel planning, and essential daily tasks. The dataset emphasizes critical mobile-specific metrics like inference latency, energy consumption, memory usage, and response quality, offering comprehensive insights into model performance under mobile constraints. Moreover, it prioritizes privacy and adaptability, assessing models' ability to perform on-device processing, maintain user privacy, and adapt to personalized usage patterns. The **Mobile-MMLU** family offers a standardized framework for developing and comparing mobile-optimized LLMs, enabling advancements in productiv-

---

*. Joint first author & equal contribution.

†. Corresponding author & project lead.

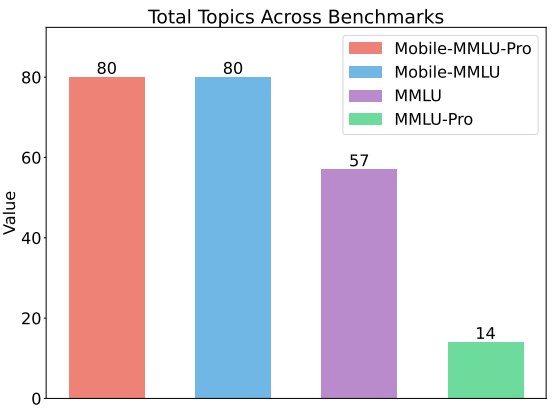

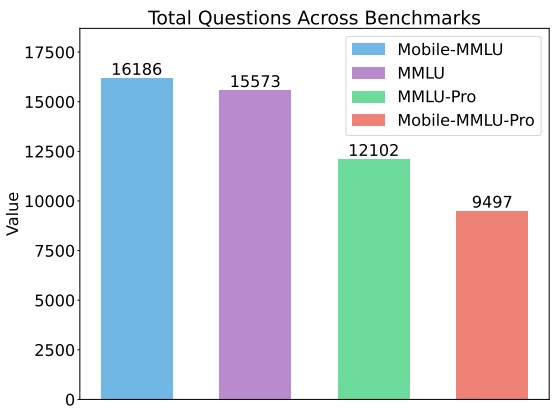

(a) Number of topics across benchmark   (b) Number of questions across benchmark

Figure 1: Statistical comparison across benchmarks. Our Mobile-MMLU and Mobile-MMLU-Pro both maintain comprehensive coverage with 80 topics each, significantly more than MMLU (57 topics) and MMLU-Pro (14 topics). In terms of question volume, Mobile-MMLU leads with 16,186 questions, followed by MMLU (15,573), MMLU-Pro (12,102), and Mobile-MMLU-Pro (9,497). Our full version contains the largest number of questions, designed for the mobile-centric evaluation of mobile-level LLMs. Our Pro version has fewer but more challenging questions, making it ideal for quick testing of strong models.

ity and decision-making within mobile computing environments. Our data is available at: `https://huggingface.co/datasets/MBZUAI-LLM/Mobile-MMLU`.

**Keywords:** Mobile-MMLU, Mobile-MMLU-Pro, LLM evaluation

## 1 Introduction

Over the past decade, the rapid adoption of mobile devices has transformed how people consume information and interact with technology (Ohme et al., 2022; Chen et al., 2015; Peltonen et al., 2018; Olson et al., 2022). As mobile hardware becomes increasingly powerful and ubiquitous, there is growing demand for on-device artificial intelligence solutions capable of supporting real-time language understanding (Zheng et al., 2025; Deng et al., 2020; Market.us, 2024). From virtual assistants to language translation apps, mobile language intelligence is reshaping communication, learning, and productivity at unprecedented scales. Despite these advancements, there remains a pressing need for benchmarks that can rigorously evaluate language understanding models specifically designed for mobile environments. Tech companies like Apple recently introduced initiatives of *Apple Intelligence* (Apple, 2024). However, their evaluation datasets and benchmarks remain focused on desktop and server-side use cases, creating a gap in the ability to assess and optimize LLMs for on-device applications.

Generally, these existing large-scale desktop or cloud-based systems-level language understanding benchmarks often fail to capture the unique constraints and performance targets required for mobile devices. Concerns like limited memory, strict power budgets, and real-time inference demands mean that a model's performance in the cloud may not translate well to the phone in your pocket. Furthermore, mobile devices exhibit diverse operating

conditions, varying hardware capabilities, network connectivity, and sensor inputs, which complicate attempts to directly apply traditional benchmarks (Wang et al., 2025; Zheng et al., 2025; Deng et al., 2020).

To address these challenges, we introduce Mobile-MMLU, a comprehensive benchmark designed to evaluate LLM performance in mobile-specific contexts. Mobile-MMLU spans 80 diverse mobile-related domains, containing over 16,000 questions carefully curated to reflect real-world mobile usage patterns. *By focusing on tasks that align with typical mobile interactions, Mobile-MMLU ensures that models are tested in scenarios that matter most to mobile users.* Moreover, our benchmark provides a standardized framework to measure mobile-specific and order-invariant[1] performance metrics, enabling developers to optimize LLMs for the unique constraints of mobile computing.

Furthermore, currently testing results often vary across multiple evaluations of the same LLM or among different strong LLMs. To address these inconsistencies and increase benchmark difficulty, we propose creating a model-consistent version of Mobile-MMLU-Pro derived from a carefully filtered subset of Mobile-MMLU (the full set). This scheme reduces test samples, lowers computational demands, and excludes questions that yield inconsistent predictions across various LLMs. Consequently, this strategy enhances test difficulty and ensures better reliability and consistency of the evaluation process.

Mobile-MMLU and Mobile-MMLU-Pro represent a significant step forward in the development and evaluation of mobile-optimized LLMs. By providing a robust and comprehensive framework, it empowers researchers and developers to address the unique challenges of mobile platforms. In sum, Mobile-MMLU family sets out to bridge the gap between state-of-the-art language models and the demands of mobile ecosystems. By providing a standardized, realistic, and rigorous testing suite, our benchmark enables the community to build, evaluate, and refine mobile-centric natural language understanding models. We envision that Mobile-MMLU family will serve as a valuable stepping stone for researchers looking to optimize the performance, efficiency, and reliability of language understanding technologies on the ever-growing landscape of mobile platforms.

## 2 Related Work

MMLU (Hendrycks et al., 2020) serves as the foundational evaluation benchmark for most LLMs, assessing their capabilities across different domains including STEM, humanities, social sciences. MMLU-Pro (Wang et al., 2024b), extending this framework by incorporating reasoning-focused questions and expanding the choice set from four to ten options. Benchmarks including HELM (Liang et al., 2022), GLUE (Wang, 2018), SuperGLUE (Wang et al., 2019), BigBench (Srivastava et al., 2022), HellaSwag (Zellers et al., 2019), GPQA (Rein et al., 2023), and ARC (Clark et al., 2018) have further contributed to evaluating LLMs' generalization and reasoning capabilities. Platforms such as OpenCompass (Contributors, 2023), Chatbot Arena (Chiang et al., 2024), and Open-LLM-Leaderboard (Myrzakhan et al., 2024) have standardized evaluation methodologies and enabled transparent

---

1. Our *order-invariance* is demonstrated through two key aspects: 1) The lengths of correct and incorrect options are maintained consistently, with the incorrect options sometimes being longer than the correct ones to prevent bias and favor in selection by the small-scale LLMs; and 2) The results remain consistent with accuracy obtained from a random-order arrangement scheme.

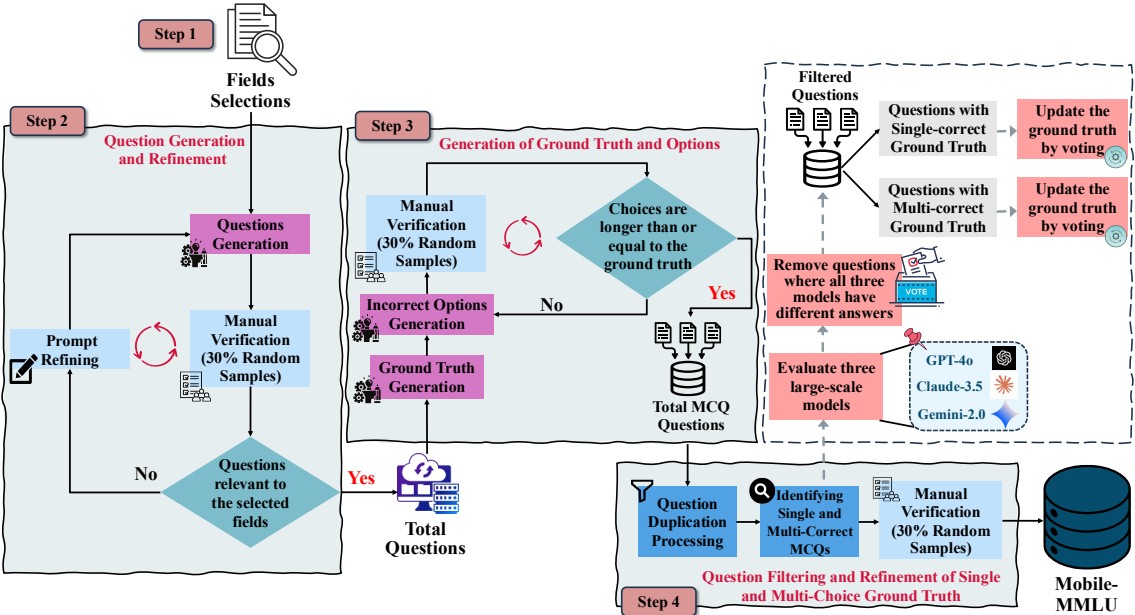

Figure 2: Data construction pipeline for Mobile-MMLU, where the left subfigure (Step 2) illustrates handling ground truth and multiple-choice questions (MCQs), and the right subfigure (Step 4) demonstrates handling multi-correct situations. The whole process includes field selection, question structuring, and iterative verification for mobile-relevant questions.

model comparisons. However, these benchmarks are primarily designed for evaluating general understanding of language models in desktop environments, overlooking the unique constraints and use cases specific to mobile platforms.

On the other hand, most of the mobile-device specific benchmarks primarily focus on technical performance metrics, overlooking the distinct nature of mobile interactions (Wang et al., 2024a; Murthy et al., 2024; Li et al., 2024b). HammerBench (Wang et al., 2024a) focuses on evaluating function-calling capabilities in real-world mobile scenarios, particularly analyzing how LLMs handle imperfect instructions and context shifts in human-LLM interactions. MobileAIBench (Murthy et al., 2024) provides a comprehensive framework for assessing both LLMs and LMMs across different model sizes and quantization levels, with particular attention to trust, safety, and hardware resource utilization on mobile devices. PalmBench (Li et al., 2024b) specifically targets the evaluation of compressed models on mobile platforms, examining the critical balance between generation, latency, and resource efficiency. Other recent work has focused on developing benchmarks for autonomous Android agents. AndroidControl (Li et al., 2024a) introduces a dataset of 15,283 demonstrations across 833 Android apps for training UI control agents. AndroidLab (Xu et al., 2024) provides a systematic framework with 138 tasks across nine apps for training and evaluating both LLMs and LMMs as Android agents (Xu et al., 2024) and AndroidWorld (Rawles et al., 2024) creates a dynamic environment with 116 programmatic tasks across 20 real-world Android apps that can generate unlimited parameterized task variations. These benchmarks evaluate models' capabilities to autonomously control Android interfaces and complete real-world mobile tasks through direct interaction with app interfaces.

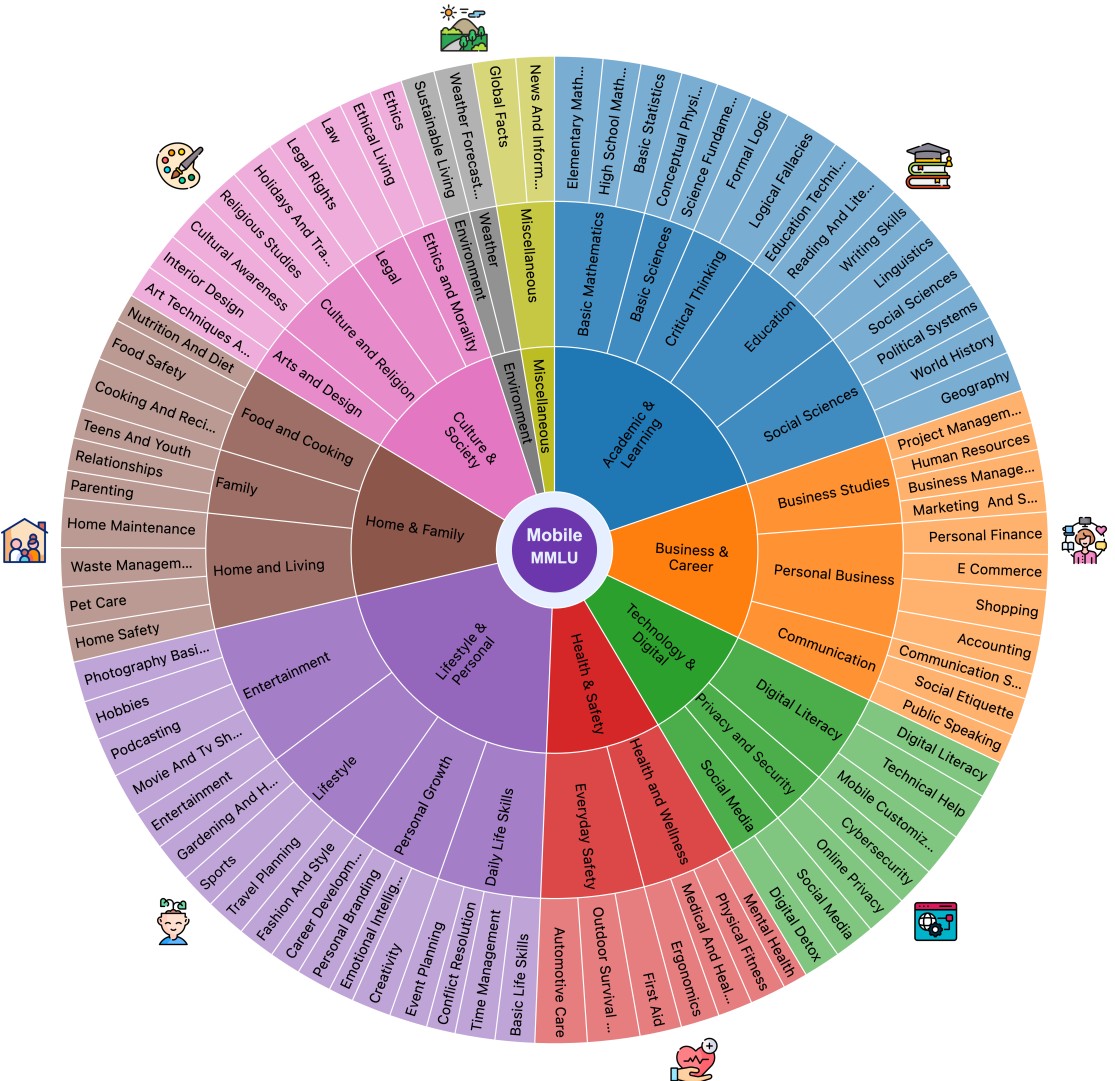

Figure 3: Illustration of topic hierarchy of Mobile-MMLU. Our benchmark consists of topics and questions related to daily life use cases like `Travel-Planning`, `First-Aid`, `Parenting`.

While these benchmarks have made significant progress in evaluating specific aspects of mobile LLM deployment, there remains a need for a unified benchmark that comprehensively assesses both the technical performance and real-world utility of LLMs in mobile-specific scenarios. Recent studies on mobile information needs (Khaokaew et al., 2023; Guy, 2016; Trippas et al., 2024) have shown that mobile users exhibit distinct interaction patterns and requirements compared to desktop users. Our work, Mobile-MMLU family, addresses this gap by providing a holistic evaluation framework that considers both the unique constraints of mobile platforms and the distinct patterns of mobile user interactions by incorporating different aspects of daily life scenarios in our benchmark.

## 3 Mobile-MMLU Benchmark

### 3.1 Overview

Our Mobile-MMLU benchmark series is designed to evaluate language models in mobile-specific contexts with two specialized versions: full Mobile-MMLU (16,186 questions) and

Mobile-MMLU-Pro (9,497 questions). Covering 80 practical domains such as *First Aid* and *Travel Planning*, as illustrated in Figure 3, these datasets prioritize real-world mobile applications for mobile-level LLMs over traditional academic subjects. This release focuses on the English scenario. All questions, answer options, and ground-truth answers are written and validated in English. A small subset, approximately 3.5%, is time sensitive (e.g., app behaviors, population counts); these items are explicitly tagged and carry a `last_verified` timestamp, and we maintain them under a versioned update policy (see Section 3.2).

Our dataset construction pipeline follows four key stages:

- *Field selection.* We aim to identify mobile-relevant domains using sources like Wiki-How, Stack Exchange, Reddit, other forums, and LLM suggestions.

- *Question / Choice generation.* We create scenario-based MCQs using GPT-4O (Hurst et al., 2024), O1-preview (Jaech et al., 2024), and human verification.

- *Similarity filtering.* We apply MPNet embeddings (Song et al., 2020) to remove duplicates (cosine similarity > 0.98).

- *Human-AI collaborative verification.* We further refine through iterative annotation and multi-model consensus validation to remove low-quality or easy question samples.

Some observed issues and precautions in our data construction:

- *LLM selection order bias.* To verify whether LLMs have a tendency to prefer certain answer positions, we test model performance by placing the ground truth (GT) answer in different ranking positions.

- *LLM bias toward option length.* We observe that LLMs (especially mobile-level LLMs) tend to select the longest option more frequently. To mitigate this, we adjust the length of incorrect options to be similar to or longer than the GT answer. This ensures that the model is selecting based on knowledge rather than guessing based on option length.

## 3.2 Dataset Maintenance and Versioning

Based on manual verification, the dataset contains approximately 3.5% time-sensitive questions. These items will be explicitly tagged and will carry a `last_verified` timestamp. We will publish regular, versioned releases (e.g., v1.1 and v1.2) with public changelogs documenting additions, removals, and any answer updates. If lightweight automated checks or community reports indicate material drift before the annual cycle, we will issue a targeted minor patch (e.g., v1.1.x). We will also enable community suggestions and corrections via Hugging Face and GitHub issues. For a broader discussion, see Section 6.

## 3.3 Benchmark Construction Pipeline

**Field Selection:** As shown in Figure 2, our benchmark construction begins with a comprehensive search to identify domains relevant to everyday activities, work, shopping, gaming, travel, and other practical scenarios on mobile devices. The objective is to ensure that the

selected fields reflect real-world user needs on mobile devices and align with the types of questions and searches that people typically make on their phones.

To achieve this, we follow a two-stage process: In the first stage, we collect an initial list of fields from a range of public sources, including Wikipedia category trees, e.g., Everyday life, Household knowledge (wikipedia), and topic groupings from popular Q&A platforms such as Quora and Reddit. To ensure broad coverage, we also leverage LLMs to suggest additional relevant fields. This multi-source approach allows us to capture both structured knowledge and user-generated interest areas, resulting in a comprehensive and inclusive set of domains. In the second stage, we ensure the mobile relevance of each field by drawing on insights from prior HCI research on mobile information-seeking behavior (Kamvar and Baluja, 2006; Church and Smyth, 2009; Aliannejadi et al., 2019). These studies show that mobile interactions are typically context-aware and occur frequently while users are in transit or multitasking. For example, users often search for information related to tasks like cooking, navigation, or how-to instructions.

Guided by these findings, we apply a qualitative criterion to assess the mobile relevance of each field: a domain is considered mobile-relevant if it commonly supports quick, practical tasks that users are likely to perform via mobile devices. To validate this selection, two annotators independently review each field and answer the question: "Is this field commonly associated with real-world mobile use?" Fields are retained if both annotators agreed. Examples of retained domains include Travel Planning, First Aid, Cooking and Recipes, and Home Maintenance.

**Question Generation:** Users in real life encounter a wide range of situations, from simple, direct queries to complex, multi-step problem-solving scenarios. To reflect this diversity, we design our dataset with questions at two difficulty levels. For each selected field, the initial questions are generated using GPT-4o and o1-preview. The first level is the ordinary questions that involve straightforward scenarios, such as "How can I check who viewed my LinkedIn profile?" or "How can I remove oil stains from my driveway?"; and the second level is the complex, scenario-based questions that require multistep reasoning, decision-making, and the ability to evaluate multiple factors before arriving at a conclusion. This comprises 6,020 questions out of the total. For instance, "I scheduled a series of posts on Facebook and Instagram using a third-party app, but some posts didn't publish. Considering potential issues like API limitations, platform policies, and app permissions, what might be causing this, and how can I fix it?" or "My car's check engine light just turned on, but I need to make a long drive tomorrow. How can I determine if it's safe to drive or if I need immediate repairs?" These two levels of questions are generated in separate batches, each following the methodology described in Step 2 of Figure 2. We apply two types of prompt formulations for the practical implementation.

**Generation of Ground Truth and Options:** The next step is to generate the correct (ground truth) answer for each question, followed by generating incorrect options. To avoid LLM bias toward option length, these incorrect options are designed to be either the same length as the correct answer or slightly longer, with differences only in specific keywords. This principle was adopted after human verification revealed that evaluated models tend to prefer longer answers. To mitigate this bias, incorrect options are designed to be equal to or longer than the correct answers while maintaining similar wording and structure. This approach not only addresses length bias but also increases the difficulty of our benchmark.

**Similarity filtering:** Once the questions with their options are constructed, a cosine similarity metric is applied across all questions to detect and remove duplicates or questions with significant overlap. This step ensures that the dataset consists of unique, non-repetitive questions. To compute cosine similarity, we use vector representations generated by the "all-mpnet-base-v2" model (Song et al., 2020) from Sentence-Transformer (Reimers and Gurevych, 2019). This model encodes each question into a dense vector of size 768, where each dimension captures semantic and contextual information about the question. The cosine similarity between two questions is then calculated as:

$$\text{Cosine Similarity} = \frac{Q_1 \cdot Q_2}{\|Q_1\| \cdot \|Q_2\|} = \frac{\sum_{k=1}^{n} Q_{1,k} Q_{2,k}}{\sqrt{\sum_{k=1}^{n} Q_{1,k}^2} \cdot \sqrt{\sum_{k=1}^{n} Q_{2,k}^2}} \tag{1}$$

where $Q_1$ and $Q_2$ represent the vectorized forms of the two questions, and $n = 768$ corresponds to the embedding dimension.

### 3.4 Human Annotation and Refinement

**Refinement During the Generation Process:** The human annotation process starts during question generation to ensure quality and relevance. It consists of two key phases:

- Phase 1: *Verification of question relevance.* Human reviewers assess whether each generated question is relevant to its field and appropriate for mobile use cases. A question is deemed mobile-relevant if it satisfies both of the following criteria:

  (1) **Mobile-context compatibility:** The question is likely to arise in real-world situations where people typically use mobile devices, such as while commuting, cooking, shopping, or multitasking.

  (2) **Mobile-task alignment:** The question reflects a practical need that users would naturally seek help with on a mobile device, such as a time-sensitive query, a task-oriented instruction, or a quick decision-making scenario.

  If a sample batch contains questions that are irrelevant or misaligned with mobile use cases, the batch is discarded and regenerated through prompt refinement.

- Phase 2: *Validation of incorrect options.* Reviewers verify that the incorrect options are indeed incorrect and ensure that they are either longer than or equal in length to the ground truth answers.

The above process is repeated for each generated batch.

**Refinement of Single and Multi-Correct Ground Truth**: We observe that some questions have multiple correct answers due to the nature of the options generated in Phase 2. To address this, as illustrated in Figure 2 (Step 4.2), we analyze responses from three large-scale models, GPT-4o, Claude-3.5, and Gemini-2.0. If all three models provide different answers (i.e., all three models' predictions are inconsistent and no two models select the same option), we remove these questions, as this suggests that multiple options are likely correct, or no absolutely correct choice here.

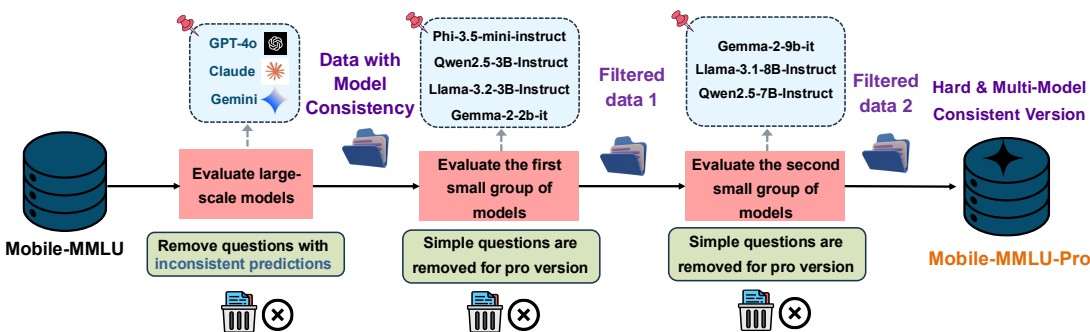

Figure 4: Data creation pipeline for Mobile-MMLU-Pro.

For the remaining questions, we update the ground truth using a voting process, where the updated ground truth corresponds to the answer agreed upon by at least two out of three models. As a result, *5.8% of the updated questions now have more than one correct choice as the ground truth*[2]. For the rest, if the voting process confirms the initial answer, the ground truth remains unchanged. Otherwise, the most agreed-upon answer is selected as the new ground truth.

## 4 Mobile-MMLU-Pro: A Multi-model Consistent and Challenging Version

Currently, most existing multiple-choice question benchmarks share a common issue: they produce inconsistent performance results when evaluated repeatedly across various runs or different strong LLMs. To mitigate this variance, we design our dataset specifically for mobile scenarios and models, aiming for consistent results on server-side models as well, minimizing variance. Moreover, we seek to increase the difficulty level and reduce testing overhead by decreasing the number of samples. To achieve these goals, we implement two steps: (1) filtering out questions consistently answered correctly by two groups of mobile-level LLMs, and (2) removing questions where predictions differ among the strongest LLMs, including GPT-4o, Claude-3.5, and Gemini-2.0.

Specifically, Mobile-MMLU-Pro is a subset of Mobile-MMLU, which is created by evaluating Mobile-MMLU on two groups of models: a small-scale-model ensemble (Qwen-3B, LLaMA-3.2 3B, Gemma-2 2B, and Phi-3.5) and a moderate-scale-model ensemble (Qwen-7B, LLaMA-8B, and Gemma-9B) (dual-model evaluation approach). Drawing inspiration from recent work in rejection sampling for language models (Apple, 2024; Yuan et al., 2023; Yang et al., 2024), we develop a two-phase evaluation framework for Mobile-MMLU-Pro that combines model-based filtering with selective rejection sampling. A consistency constraint is further applied by removing questions where predictions differ among the strongest models of GPT-4o, Claude-3.5, and Gemini-2.0.

Our approach differs from previous rejection sampling methods in several key aspects. While Yuan et al. (Yuan et al., 2023) focus on using rejection sampling to identify correct reasoning paths for mathematical problems, and Apple (Apple, 2024) employs a teacher committee (iTeC) for iterative refinement, our methodology specifically targets the identifi-

---

2. This means our benchmark contains 5.8% questions with multi-choice as ground-truth.

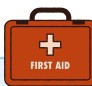

**First Aid**

**Question:**

**Someone is bleeding profusely from their leg after an accident; I applied pressure but it's not stopping; should I use a tourniquet, and how?**

**Options:**

A. Yes, in cases of excessive bleeding that does not cease with direct pressure, employing a tourniquet could be an option to manage the bleeding risk. To use a tourniquet, position it below the bleeding site, ideally 2-3 inches below the wound but avoiding placement on joints like the shoulder or wrist. Ensure it is tightened until bleeding is significantly reduced and it is secured properly. Document the exact time of application as this timing is important for healthcare workers to understand how long it has been used. A tourniquet is meant for use in dire, life-endangering situations, and it is imperative to get professional medical aid as soon as achievable.

B. Yes, when facing severe blood loss that cannot be controlled with direct pressure, using a tourniquet might be an effective method to slow down the bleeding. To deploy a tourniquet, place it below the site of the bleeding injury, ideally 2-3 inches away from the wound but not over a joint such as an ankle or wrist. Tighten the device so the bleeding slows significantly and make sure it is fastened securely. It is crucial to record the time the tourniquet was applied, as medical personnel need to be aware of the duration it has been applied. Only use a tourniquet in critical emergencies, and ensure to seek professional medical assistance without delay.

C. Yes, in situations where there is major bleeding that does not stop despite applying direct pressure, applying a tourniquet might be considered a method to attempt controlling the bleeding. To use a tourniquet, place it directly on the site of the bleeding, ideally wrapping it tightly around the wound itself. Avoid placing it on joints such as the knee or elbow, and secure it firmly. Remember to note the exact time when the tourniquet was applied, as medical staff will need this information to assess the duration it has been in place. A tourniquet should only be employed in critical, life-threatening situations, and it is crucial to contact medical professionals immediately for further assistance.

D. Yes, in situations where there is severe bleeding that does not stop with direct pressure, applying a tourniquet can be an effective method to control the bleeding. To use a tourniquet, place it above the site of the bleeding, ideally 2-3 inches above the wound but not on a joint like the knee or elbow. Tighten it until the bleeding stops and secure it in place. Note the time the tourniquet was applied, as it is important for medical personnel to know how long it has been in place. A tourniquet should only be used in life-threatening situations, and professional medical help should be sought as soon as possible. ✅

Figure 5: Example question from the *First Aid* field in Mobile-MMLU dataset.

cation of discriminative questions that can effectively differentiate model capabilities. This is achieved through a carefully orchestrated combination of multiple model-based evaluation and selective rejection sampling. The resulting Mobile-MMLU-Pro benchmark demonstrates strong discriminative power and consistency across different model scales while maintaining focus on practical, mobile-relevant scenarios.

## 5 Experiments

### 5.1 Setup

**Models.** To comprehensively evaluate model performance on Mobile-MMLU and Mobile-MMLU-Pro, we select a diverse set of state-of-the-art LLMs ranging from 1B to 9B parameters. Our model suite includes Gemma-2-9B-it (Team et al., 2024), Qwen2.5-7B-instruct (Yang et al., 2024), Llama-3.1-8B-instruct (Dubey et al., 2024), Qwen2.5-3B-instruct (Yang et al., 2024), Phi-3.5-mini-instruct (Abdin et al., 2024), Llama-3.2-3B-instruct (Meta, 2024), Gemma-2-2B-it (Team et al., 2024), Ministral-8B-instruct, Qwen2.5-1.5B-instruct (Yang et al., 2024) and Llama-3.2-1B-instruct (Meta, 2024). This selection encompasses models of varying architectures and parameter counts, enabling a thorough analysis of the relationship between model size and performance on mobile-oriented tasks.

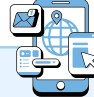

**Social Media**

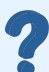

**Question:**

**How do I share a post on Instagram's 'Close Friends' list?**

**Options:**

**A. To share a post on Instagram's 'Close Friends' list, you should create a post and choose the 'Close Friends' option after you have posted it to your story.**

**B. To share a post on Instagram's 'Close Friends' list, you need to create a story and select the 'Close Friends' option before posting.** ✓

**C. To share a post on Instagram's 'Close Friends' list, you need to create a story and then select the 'Friends Only' option before you proceed with posting it.**

**D. To share a post with Instagram's 'Close Friends' list, you must first create a story and then choose the 'Favorites' option prior to posting the story.**

Figure 6: Example question from the *Social Media* field in Mobile-MMLU dataset.

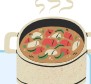

**cooking and recipes**



**Question:**

**I accidentally added baking powder instead of baking soda to my cookie dough; how will this affect the cookies, and can I adjust the dough to fix it?**

**Options:**

**A. Using baking powder instead of baking soda in your cookie dough will result in a different texture and possibly flavor. Baking powder contains both an acid and a base, so it won't react with the dough in the same way as baking soda, which requires an acidic ingredient to activate. This means your cookies might be less spread out and puffier, with a lighter texture. They also might have a slightly tangier taste due to the acid in baking powder. To adjust the dough, you can try adding a small amount of cream of tartar or lemon juice to activate any remaining baking soda if present, or you can simply proceed with the dough as is, keeping in mind the texture and flavor changes.** ✓

**B. Using baking powder instead of baking soda in your cookie dough will result in a different texture and possibly flavor. Baking powder contains both a base and a stabilizer, so it won't react with the dough in the same way as baking soda, which requires a specific acidic compound to activate. This substitution means your cookies might be more spread out and softer, with a moister texture. They also might have a slightly saltier taste due to the stabilizer in baking powder. To adjust the dough, you can try adding a small amount of yogurt or molasses to complement any changes in taste, or you can proceed with the dough as is, keeping in mind the textural and flavor modifications.**

**C. Using baking powder instead of baking soda in your cookie dough will result in a different texture and possibly flavor. Baking powder contains both a neutralizer and a base, so it won't react with the dough in the same way as baking soda, which requires a particular acidic reactant to activate. This means your cookies might be less fluffy and crispier, with a denser texture. They also might have a slightly bitter taste due to the neutralizer in baking powder. To adjust the dough, you can try adding a small amount of apple cider vinegar or honey to mask any undesired flavors, or you can simply go ahead with the dough as is, keeping in mind the texture and flavor shifts.**

**D. Using baking powder instead of baking soda in your cookie dough will result in a different texture and possibly flavor. Baking powder contains both a neutralizer and a base, so it won't react with the dough in the same way as baking soda, which requires a particular acidic reactant to activate. This means your cookies might be less fluffy and crispier, with a denser texture. They also might have a slightly bitter taste due to the neutralizer in baking powder. To adjust the dough, you can try adding a small amount of apple cider vinegar or honey to mask any undesired flavors, or you can simply go ahead with the dough as is, keeping in mind the texture and flavor shifts.**

Figure 7: Example question from the *Cooking and Recipes* field in Mobile-MMLU dataset.

**Evaluation.** For our evaluation framework, we use **lm-eval-harness** (Gao et al., 2024) to assess the performance of the models. Given that Mobile-MMLU and Mobile-MMLU-Pro consist entirely of multiple-choice questions, we focus on accuracy as our primary evaluation

metric. This approach allows for objective comparison across models while maintaining consistency with existing benchmarking practices in the field. For the Open-style Question (OSQ) setting, we follow the multi-stage framework of Myrzakhan et al. (2024). We first apply a two-step filtering process. First, we classify each MCQ as convertible ("YES") or non-convertible ("NO") using a task-specific prompt that outputs only the verdict. Second, for items classified as "NO", we assign a confidence score from 1 to 10 indicating the likelihood of successful open-style conversion, and set a threshold of 5. We discard "NO" items with scores below 5, while moving "NO" items with a score of 5 or higher into the convertible pool together with all "YES" cases. After conversion, we evaluate open-style answers with an LLM judge using the verification prompt of Myrzakhan et al. (2024). Given the question, the model's answer, and the MCQ ground-truth answer, the judge returns only "Correct" or "Incorrect", and the prompt instructs it to accept extra non-conflicting information, mark conflicts with the ground truth as "Incorrect", and ignore minor differences in phrasing and punctuation.

## 5.2 Main Results

**Overall Performance Comparison.** Our evaluation reveals several significant patterns across different model scales and architectures. Table 2 presents the zero-shot performance of both mobile-friendly and general-purpose (slightly larger size) models across MMLU, MMLU-Pro, Mobile-MMLU, and Mobile-MMLU-Pro benchmarks. On our benchmark, the performance gap between the strongest and weakest models is larger compared to other benchmarks, highlighting the greater discriminative capability of our dataset, high-performing models achieve better scores, while weaker models perform worse. For instance, the lowest-performing model, Nemotron-Mini-4B-Instruct, scores 35.1% on our dataset but achieves 56.8% on MMLU. Conversely, the highest-performing model, Qwen2.5-3B-Instruct, obtains a higher score of 68.1% on our benchmark, surpassing its 65.4% score on MMLU. We emphasize that a strong benchmark dataset should not aim merely to lower model performance scores, rather, it should accentuate distinctions between models, clearly highlighting significant performance gaps and enabling better differentiation of their capabilities. In addition, our results demonstrate that strong performance on traditional benchmarks does not necessarily translate to superior performance on mobile-specific tasks. A notable example is Phi-3.5-mini-instruct, which achieves impressive scores on MMLU (68.7%) but shows relatively lower performance on Mobile-MMLU (63.7%). Conversely, Qwen2.5-3B-Instruct, despite its modest performance on MMLU (65.4%), excels on Mobile-MMLU (68.1%), even outperforming some larger 8B parameter models.

**Open-Style Question Answering.** Open-Style Questions (OSQ) are a stricter setting in which models must produce the answer directly without being guided by multiple-choice options. This format places greater demands on recall, reasoning, and answer precision compared to standard MCQ-based evaluation. Performance across models varies widely, as shown in Table 3. In Mobile-MMLU OSQ, the lowest score is 13.7% (Nemotron-Mini-4B-Instruct) and the highest is 54.6% (Gemma-2-9b-it), a gap exceeding 40 percentage points. This spread is larger than what is often seen in traditional benchmarks, suggesting that open-ended mobile-oriented evaluation surfaces sharper capability distinctions between systems. The results also show that model size is not the sole determinant of success. Qwen2.5-3B-Instruct achieves 50.9% on Mobile-MMLU OSQ and 38.6% on Mobile-MMLU-

| Model | Size | Params | Throughput (token/s) | | Peak RSS (GiB) | |
|---|---|---|---|---|---|---|
| | | | **Prefill 512** | **Gen 128** | **Prefill 512** | **Gen 128** |
| **iPhone 14** | | | | | | |
| Llama-3.2-1B-Instruct | 0.95 GiB | 1.24 B | 433.17 ± 23.42 | 24.64 ± 12.64 | 1.14 | 1.14 |
| Qwen2.5-1.5B-Instruct | 1.09 GiB | 1.54 B | 350.94 ± 27.26 | 25.45 ± 0.13 | 1.33 | 1.34 |
| Gemma-2-2b-it | 2.04 GiB | 2.61 B | 213.78 ± 30.10 | 15.39 ± 0.36 | 2.30 | 2.29 |
| Qwen2.5-3B-Instruct | 2.03 GiB | 3.09 B | 118.26 ± 84.97 | 13.63 ± 0.48 | 2.21 | 2.21 |
| Llama-3.2-3B-Instruct | 2.17 GiB | 3.21 B | 148.44 ± 20.70 | 13.39 ± 0.17 | 2.29 | 2.29 |
| Phi-3.5-mini-instruct | 2.23 GiB | 3.82 B | 110.15 ± 14.89 | 11.24 ± 0.21 | 2.83 | 2.83 |
| **Samsung S25 Ultra (Android)** | | | | | | |
| Llama-3.2-1B-Instruct | 0.95 GiB | 1.24 B | 98.34 ± 3.39 | 52.34 ± 2.40 | 0.92 | 0.92 |
| Qwen2.5-1.5B-Instruct | 1.09 GiB | 1.54 B | 66.31 ± 0.68 | 39.05 ± 1.22 | 1.07 | 1.07 |
| Gemma-2-2b-it | 2.04 GiB | 2.61 B | 38.82 ± 5.23 | 14.45 ± 4.02 | 1.80 | 1.80 |
| Qwen2.5-3B-Instruct | 2.03 GiB | 3.09 B | 24.80 ± 4.11 | 13.84 ± 3.40 | 1.96 | 1.96 |
| Llama-3.2-3B-Instruct | 2.17 GiB | 3.21 B | 24.18 ± 2.31 | 13.32 ± 2.55 | 2.08 | 2.08 |
| Phi-3.5-mini-instruct | 2.23 GiB | 3.82 B | 17.20 ± 1.97 | 11.57 ± 3.33 | 2.53 | 2.53 |
| **NVIDIA Jetson Orin NX** | | | | | | |
| Llama-3.2-1B-Instruct | 0.95 GiB | 1.24 B | 1549.9 ± 30.0 | 45.8 ± 1.0 | 0.95 | 0.95 |
| Qwen2.5-1.5B-Instruct | 1.09 GiB | 1.54 B | 1311.5 ± 74.8 | 32.9 ± 1.6 | 1.11 | 1.11 |
| Gemma-2-2b-it | 2.04 GiB | 2.61 B | 1052.8 ± 6.4 | 23.3 ± 0.2 | 1.78 | 1.78 |
| Qwen2.5-3B-Instruct | 2.03 GiB | 3.09 B | 778.9 ± 11.0 | 21.7 ± 0.1 | 1.99 | 1.99 |
| Llama-3.2-3B-Instruct | 2.17 GiB | 3.21 B | 776.2 ± 9.3 | 20.5 ± 0.2 | 2.08 | 2.08 |
| Phi-3.5-mini-instruct | 2.23 GiB | 3.82 B | 576.5 ± 4.6 | 19.1 ± 0.2 | 2.37 | 2.37 |
| **Raspberry Pi 5** | | | | | | |
| Llama-3.2-1B-Instruct | 0.95 GiB | 1.24 B | 32.21 ± 1.41 | 11.54 ± 0.18 | 0.89 | 0.89 |
| Qwen2.5-1.5B-Instruct | 1.09 GiB | 1.54 B | 22.11 ± 0.26 | 9.29 ± 0.04 | 1.04 | 1.04 |
| Gemma-2-2b-it | 2.04 GiB | 2.61 B | 13.97 ± 0.16 | 4.66 ± 0.00 | 1.76 | 1.76 |
| Qwen2.5-3B-Instruct | 2.03 GiB | 3.09 B | 10.35 ± 0.04 | 4.15 ± 0.00 | 1.94 | 1.94 |
| Llama-3.2-3B-Instruct | 2.17 GiB | 3.21 B | 10.13 ± 0.21 | 4.33 ± 0.00 | 2.06 | 2.06 |
| Phi-3.5-mini-instruct | 2.23 GiB | 3.82 B | 7.07 ± 0.13 | 3.55 ± 0.01 | 2.51 | 2.52 |

Table 1: Cross-device on-device inference performance of Q4_K_M–quantized small-scale LLMs using `llama.cpp` on iPhone 14, Samsung S25 Ultra (Android), NVIDIA Jetson Orin NX, and Raspberry Pi 5. Throughput is reported for prompt prefilling of 512 tokens (Prefill 512) and autoregressive generation of 128 tokens (Gen 128) as mean ± std (tokens/s). Peak resident set size (RSS) is reported separately for prefilling and generation. *Size* denotes the on-disk model size (GiB) and *Params* the parameter count. All models fall in the 1B–3.8B parameter range, representative of mobile-friendly deployments.

Pro OSQ comparable to or better than certain 7B–8B models. Similarly, Phi-3.5-mini-instruct stands out among compact models, reaching 48.4% OSQ accuracy, outperforming some larger competitors. Mobile-MMLU-Pro OSQ variant is more challenging, with uniformly lower scores. High-performing models tend to preserve their relative ranking, but the performance gap between top-tier and mid-tier models becomes even more pronounced.

| Model | MMLU (0-shot) | MMLU-Pro (0-shot) | Mobile-MMLU (0-shot) | Mobile-MMLU-Pro (0-shot) |
|---|---|---|---|---|
| **Mobile-Friendly Models** | | | | |
| Nemotron-Mini-4B-Instruct | 56.8 | 18.1 | 35.1 | 30.8 |
| Llama-3.2-1B-Instruct | 45.9 | 7.5 | 34.5 | 31.1 |
| Gemma-2-2b-it | 56.8 | 17.2 | 38.9 | 31.2 |
| Falcon3-3B-Instruct | 55.8 | 22.3 | 42.6 | 37.2 |
| Granite-3.1-3b-a800m-instruct | 50.5 | 12.8 | 43.6 | 39.4 |
| Llama-3.2-3B-Instruct | 60.3 | 24.4 | 50.2 | 42.0 |
| Granite-3.1-2b-instruct | 54.3 | 20.2 | 48.1 | 42.2 |
| Qwen2.5-1.5B-Instruct | 60.1 | 19.9 | 49.7 | 43.0 |
| Exaone-3.5-2.4B-Instruct | 58.1 | 25.3 | 53.7 | 47.7 |
| Phi-3.5-mini-instruct | 68.7 | 32.9 | 63.7 | 54.8 |
| Qwen2.5-3B-Instruct | 65.4 | 25.0 | 68.1 | 60.6 |
| **General-Purpose Models** | | | | |
| Olmo-2-1124-7B-Instruct | 59.3 | 18.6 | 49.6 | 42.9 |
| Falcon3-7B-Instruct | 68.0 | 34.3 | 52.4 | 46.8 |
| Falcon3-10B-Instruct | 71.6 | 38.1 | 54.3 | 49.1 |
| Yi-1.5-6B-Chat | 61.8 | 24.4 | 60.5 | 54.7 |
| Llama-3.1-8B-Instruct | 67.9 | 30.7 | 66.9 | 57.1 |
| Granite-3.1-8b-instruct | 64.5 | 28.2 | 60.8 | 57.7 |
| Internlm2_5-7b-chat | 67.7 | 30.4 | 64.3 | 58.6 |
| Ministral-8B-Instruct-2410 | 64.0 | 25.4 | 71.5 | 63.6 |
| Yi-1.5-9B-Chat | 68.3 | 33.1 | 72.7 | 67.7 |
| Qwen2.5-7B-Instruct | 71.7 | 36.5 | 74.9 | 68.4 |
| Gemma-2-9b-it | 71.8 | 31.9 | 75.0 | 69.1 |

Table 2: Comparison of performance on Mobile-MMLU and Mobile-MMLU-Pro with prior MMLU and MMLU-Pro benchmarks for different sizes of models.

For example, Gemma-2-9b-it scores 48.7%, while Granite-3.1-2b-instruct drops to 21.3%, widening the performance separation compared to OSQ. Overall, these results indicate that OSQ-based evaluation especially in a mobile-specific context offers a more stringent and discriminative test of model ability, revealing strengths and weaknesses that might remain hidden in MCQ-style benchmarks.

**Hardware Performance Evaluation.** We conduct hardware tests with the `llama.cpp` test framework lla (2023), using a modified SwiftUI example on iOS and the official command-line interface (CLI) on Android and Linux. These tests are performed on iPhone 14 (iOS), Samsung S25 Ultra (Android), NVIDIA Jetson Orin NX (Linux), and Raspberry Pi 5 (Linux) to evaluate on-device inference performance. All evaluated LLMs are converted to the GGUF format and quantized using the Q4_K_M method, a quantization technique recognized for its efficiency in mobile-optimized models. Tests use GPU execution on iOS via Metal app (2024) and on NVIDIA Jetson Orin NX via CUDA/cuBLAS, and CPU execution on Samsung S25 Ultra (Android) and Raspberry Pi 5, ensuring a consistent runtime setup per device. For each model, we record the on-device GGUF file size and parameter count. Notably, Q4_K_M-quantized models include additional quantization parameters (e.g., scaling factors), which result in a slightly higher parameter count compared to the original models.

| Model | Mobile-MMLU OSQ (0-shot) | Mobile-MMLU-Pro OSQ (0-shot) |
|---|---|---|
| **Mobile-Friendly Models** | | |
| Nemotron-Mini-4B-Instruct | 13.7 | 10.7 |
| Llama-3.2-1B-Instruct | 19.2 | 15.8 |
| Gemma-2-2b-it | 21.1 | 14.2 |
| Falcon3-3B-Instruct | 25.4 | 15.7 |
| Granite-3.1-3b-a800m-instruct | 21.2 | 19.0 |
| Llama-3.2-3B-Instruct | 28.4 | 24.8 |
| Granite-3.1-2b-instruct | 24.2 | 21.3 |
| Qwen2.5-1.5B-Instruct | 33.8 | 19.9 |
| Exaone-3.5-2.4B-Instruct | 34.5 | 32.6 |
| Phi-3.5-mini-instruct | 48.4 | 31.7 |
| Qwen2.5-3B-Instruct | 50.9 | 38.6 |
| **General-Purpose Models** | | |
| Olmo-2-1124-7B-Instruct | 31.2 | 24.5 |
| Falcon3-7B-Instruct | 35.8 | 30.2 |
| Falcon3-10B-Instruct | 29.7 | 24.5 |
| Yi-1.5-6B-Chat | 42.1 | 36.3 |
| Llama-3.1-8B-Instruct | 51.0 | 41.2 |
| Granite-3.1-8b-instruct | 44.8 | 41.7 |
| Internlm2_5-7b-chat | 40.8 | 35.1 |
| Ministral-8B-Instruct-2410 | 50.5 | 42.6 |
| Yi-1.5-9B-Chat | 49.6 | 44.6 |
| Qwen2.5-7B-Instruct | 52.6 | 46.1 |
| Gemma-2-9b-it | 54.6 | 48.7 |

Table 3: Comparison of model performance on Mobile-MMLU OSQ and Mobile-MMLU-Pro OSQ benchmarks. OSQ (*Open-Style Questions*) requires direct answer generation without multiple-choice options.

| Model | Ori. | A | B | C | D | $R\ \#1$ | $R\ \#2$ | $R\ \#3$ | $R\ \#4$ |
|---|---|---|---|---|---|---|---|---|---|
| Phi-3.5-mini-instruct | 63.7 | 45.77 | 70.67 | 76.99 | 87.66 | 65.61 | 65.89 | 65.05 | 65.32 |
| Qwen2.5-3B-Instruct | 68.1 | 53.74 | 79.04 | 74.30 | 83.29 | 70.22 | 69.85 | 70.36 | 69.99 |
| Llama-3.2-3B-Instruct | 50.2 | 65.17 | 61.74 | 57.12 | 56.54 | 55.82 | 54.81 | 54.64 | 54.97 |
| Llama-3.1-8B-Instruct | 66.9 | 50.78 | 88.45 | 78.94 | 75.45 | 68.75 | 68.87 | 67.45 | 67.83 |
| Qwen2.5-7B-Instruct | 74.9 | 54.82 | 85.39 | 86.23 | 88.72 | 75.91 | 75.26 | 73.43 | 74.19 |
| gemma-2-9b-it | 75.0 | 90.63 | 70.30 | 68.25 | 80.46 | 76.35 | 76.72 | 75.39 | 75.21 |
| gemma-2-2b-it | 38.9 | 67.67 | 39.40 | 41.63 | 37.37 | 42.11 | 41.74 | 40.85 | 41.43 |

Table 4: Ablation of *Selection Order Bias* using different models on Mobile-MMLU.

Two types of inference benchmarks are conducted: (*i*) **Prefilling 512**, which measures performance during the initial phase when 512 tokens are processed, and (*ii*) **Text Generation 128**, which measures performance during the generation phase when 128 tokens are produced. For each test, we record performance metrics including token throughput

| Model | Ori. | A | B | C | D | R #1 | R #2 | R #3 | R #4 |
|-------|------|------|------|------|------|------|------|------|------|
| Phi-3.5-mini-instruct | 54.8 | 41.19 | 62.78 | 66.75 | 73.94 | 57.53 | 52.74 | 55.89 | 57.11 |
| Qwen2.5-3B-Instruct | 60.6 | 53.21 | 71.22 | 68.36 | 71.28 | 63.16 | 66.12 | 65.39 | 65.62 |
| Llama-3.2-3B-Instruct | 42.0 | 54.36 | 49.96 | 50.54 | 48.88 | 47.12 | 46.38 | 44.72 | 42.69 |
| Llama-3.1-8B-Instruct | 57.1 | 48.89 | 75.18 | 66.21 | 63.45 | 54.81 | 56.11 | 56.21 | 53.87 |
| Qwen2.5-7B-Instruct | 68.4 | 54.52 | 77.55 | 76.98 | 79.14 | 71.24 | 69.29 | 69.15 | 70.37 |
| gemma-2-9b-it | 69.1 | 85.14 | 65.32 | 63.11 | 73.12 | 70.25 | 68.99 | 70.72 | 70.15 |
| gemma-2-2b-it | 31.2 | 51.65 | 34.11 | 36.77 | 29.91 | 32.21 | 33.50 | 30.76 | 31.51 |

Table 5: Ablation of *Selection Order Bias* using different models on Mobile-MMLU-Pro.

(measured in tokens per second) and peak memory usage (maximum RAM utilization). Results in Table 1 show that the smallest model, Llama-3.2-1B-Instruct, achieves the highest throughput in both benchmarks (e.g., iPhone 14: 433 tokens/s in prefilling, 24.6 tokens/s in generation). Throughput decreases with model size, with Phi-3.5-mini-instruct (3.82B) averaging 11.2 tokens/s in generation on iPhone 14 and following the same trend on other devices. Peak memory usage scales with size, yet all models fit within the memory budgets of the four devices. Notably, on iPhone 14, Qwen2.5-3B-Instruct shows high variance in prefilling throughput, possibly due to sequence initialization or scheduling effects.

### 5.3 Analysis

**Performance Distribution Analysis.** One of the most striking observations is the wide performance variance across models on Mobile-MMLU compared to other benchmarks. While the performance spread on MMLU ranges from 45.9% to 71.8%, and MMLU-Pro from 7.5% to 36.5%, Mobile-MMLU exhibits a wider relative range from 34.5% to 75.0%. This increased variance is particularly pronounced among smaller models (1-3B parameters), providing valuable insights for mobile deployment scenarios where model size constraints are critical. Furthermore, it can be seen that model size alone does not determine performance. For instance, among similarly-sized models in the 3B parameter range, we observe substantial performance differences: Qwen2.5-3B-Instruct achieves 68.1% accuracy on Mobile-MMLU, while Llama-3.2-3B-Instruct scores 50.2%, despite both models having comparable parameter counts.

**LLM Selection Order Bias.** To investigate how the position of the correct answer affects model predictions, we examine the following settings:

- *Our Original Dataset Construction Strategy*: Initially, the correct options (A/B/C/D) are randomly assigned and then filtered, resulting in a slight non-uniform distribution in the final dataset but the results are still close and stable relative to the full random-order performance (as shown in the first group of Tables 4 and 5).

- *Systematic Placement*: The correct answers are systematically rotated across A/B/C/D, while the incorrect options are randomly assigned (as shown in the middle group of Tables 4 and 5).

- *Fully Randomized Distribution*: Both correct and incorrect options are randomly distributed (as shown in the last group of Tables 4 and 5).

The *Ori.* columns in Tables 4 and 5 present the results on our **Mobile-MMLU** and **Pro** benchmarks, respectively. Our findings show strong consistency with the random-order results, exhibiting minimal variance. Thus, we refer to our dataset as *order-invariant*, reflecting its stable properties within these benchmarks. Our results in middle groups of these two tables also indicate that small LLMs are highly sensitive to the order of answer choices, with performance variance reaching to more than 10%. This highlights the importance of fair dataset construction, particularly through systematic balancing. However, the most robust solution is to adopt the open-llm-leaderboard (Myrzakhan et al., 2024) scheme, converting multiple-choice tasks into open-ended generation tasks, which eliminates this issue entirely. We plan to conduct further research in this area.

## 6 Conclusion

This paper introduced Mobile-MMLU family, a novel benchmark family designed to evaluate language models in mobile contexts, addressing a critical gap in assessing mobile-optimized language models. Through careful curation of 80 diverse topics with 16,186 questions, our Mobile-MMLU focuses on practical, mobile-relevant scenarios that better reflect real-world mobile interactions. The complementary Mobile-MMLU-Pro benchmark, created through our rigorous multi-model-consistency based rejection sampling approach, provides a more challenging and consistent evaluation set while maintaining focus on mobile-specific use cases. Both of the two benchmarks enjoy *mobile-centric* and *order-invariant* properties. Our comprehensive analysis demonstrates that Mobile-MMLU and Mobile-MMLU-Pro occupy a distinct semantic space compared to traditional benchmarks like MMLU and MMLU-Pro, with consistently higher Mobile Relevance Scores.

## Ethical Statements and Limitations

**Time-sensitive Questions in the Dataset.** Upon manual verification, our dataset contains approximately 3.5% of time-sensitive questions. These questions are included because they reflect the types of queries people commonly ask on their mobile phones, such as "What is the best application?", "What is the population of a city?", or "How do I post a photo?" Incorporating these questions is essential to creating a relevant mobile benchmark, as they represent practical, real-world use cases. While the ground truth answers for these questions are currently accurate, we acknowledge that they may become outdated over time, for instance, in 10 years, the answers might no longer be valid. To address this limitation, our benchmark will be regularly updated to maintain its relevance. See Section 3.2 for the maintenance and versioning policy. We also introduce Mobile-MMLU-Pro, a subset of our dataset that excludes simple, sensitive, and inconsistent questions. This version offers a comparable size to MMLU-Pro and provides a more stable and consistent benchmark for evaluating model performance without the complications introduced by time-sensitive or location-sensitive queries.

**Multi-choice to Open-ended Questions.** Similar to many other datasets, our benchmark primarily focuses on multi-choice questions, potentially oversimplifying the complexity of real-world language tasks encountered by mobile users. This format may not fully represent the diverse linguistic interactions and contextual complexities in mobile environments,

such as conversational nuances, multi-turn dialogue, or dynamic contextual information. Moreover, the benchmark's language diversity and domain coverage may not fully represent the global and multilingual nature of mobile user populations, restricting its comprehensive generalizability across regions and languages for different countries.

**Social Impact and Ethical Considerations.** We consider several dimensions of social impact and responsible use in designing and releasing Mobile-MMLU, following the evaluative guidance proposed by Solaiman et al. (2023). On-device inference can enhance privacy by reducing cloud transmission of user queries; however, residual privacy risks may remain via logs, caches, or crash diagnostics. Our benchmark does not use personal user data; items are LLM-generated and human-verified, and it evaluates only model outputs without collecting or processing end-user data. Several categories in our taxonomy involve higher-risk domains such as health, cybersecurity, and online privacy (Hagendorff, 2024); we caution against using benchmark results in these areas as a proxy for real-world safety or decision-making. The benchmark is currently English-only and reflects a limited pool of content sources, which may introduce representational bias or limit applicability across different cultures or populations. We also acknowledge the potential for temporal drift and maintain a transparent versioning policy to address it (see Section 3.2). While we have not observed adverse impacts from our benchmark, we offer these considerations as proactive guidance to support ethical and informed use.

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

# Appendix

## Contents

## Appendix A. Data Statistics and Distribution Analysis

Our analysis shows a significant gap in the existing general purpose benchmarks for evaluating mobile-specific LLM capabilities. While benchmarks like MMLU (Hendrycks et al., 2020) and MMLU-Pro (Wang et al., 2024b) are good for assessing broad knowledge and general abilities, they include numerous topics rarely encountered in mobile contexts, such as advanced theoretical concepts or extensive coding tasks. They underrepresent everyday mobile scenarios like quick recipe lookups, travel recommendations, or context-aware assistance.

This misalignment between benchmark content and actual mobile use cases can lead to suboptimal model selection and evaluation for mobile deployments. Figure 8, demonstrates the topic distribution across Mobile-MMLU, MMLU, and MMLU-Pro benchmarks. We use "all-mpnet-base-v2" (Song et al., 2020) model from Sentence-Transformer (Reimers and Gurevych, 2019) to get the sentence embedding for each of the questions from all three benchmark. We represent the topic using the average of all the question embedding in that topic and then use PCA (Maćkiewicz and Ratajczak, 1993) to get two dimensional representation of the topics. From the scatter plot, we can observe that Mobile-MMLU topics occupy a distinct semantic space compared to the topics of MMLU and MMLU-Pro benchmark. This clear separation in the topic distribution highlights Mobile-MMLU's unique focus on practical, mobile-relevant scenarios, complementing existing benchmarks rather than overlapping with them. The distinct clustering pattern validates Mobile-MMLU's contribution as a specialized benchmark tailored for evaluating mobile-oriented language models.

In order to further validate our hypothesis, we use GPT-4o (Achiam et al., 2023) as a judge (Zheng et al., 2023) to evaluate the Mobile Relevance Score (MRScore) of different questions in our benchmark. We then aggregate these mobile relevance scores of the questions at the topic level. Given a system prompt $S$ defining mobile expertise and evaluation guidelines, and a user prompt $C$ specifying evaluation criteria (practical value, mobile-friendliness, usage patterns), we define the MRScore for a question $q$ as:

$$\text{MRScore}(q) = f_{\text{LLM}}(q|S, C) \tag{2}$$

This process can be decomposed implicitly through $round(\lambda_1 \cdot P(q) + \lambda_2 \cdot M(q) + \lambda_3 \cdot U(q)) \in [1, 10]$, where $P(q)$, $M(q)$, and $U(q)$ represent practical value, mobile-friendliness, and usage pattern scores respectively, with $\lambda_1, \lambda_2, \lambda_3$ as implicit weights. For a topic $T$ containing $n$ questions, the aggregate score can be formulated as:

$$\text{TopicMRScore}(T) = \frac{1}{n} \sum_{q \in T} \text{MRScore}(q) \tag{3}$$

where each MRScore($q$) is conditioned on the same system and user prompts. Figures 9a and 9b show the distribution of MRScores across different benchmarks. The analysis reveals that questions in Mobile-MMLU consistently achieve higher MRScores compared to both MMLU and MMLU-Pro. The distributions show clear separation, with Mobile-MMLU questions predominantly scoring between 5 to 9, while traditional benchmarks cluster around 2 to 4. This quantitative validation confirms that our benchmark effectively captures mobile-specific use cases and scenarios.

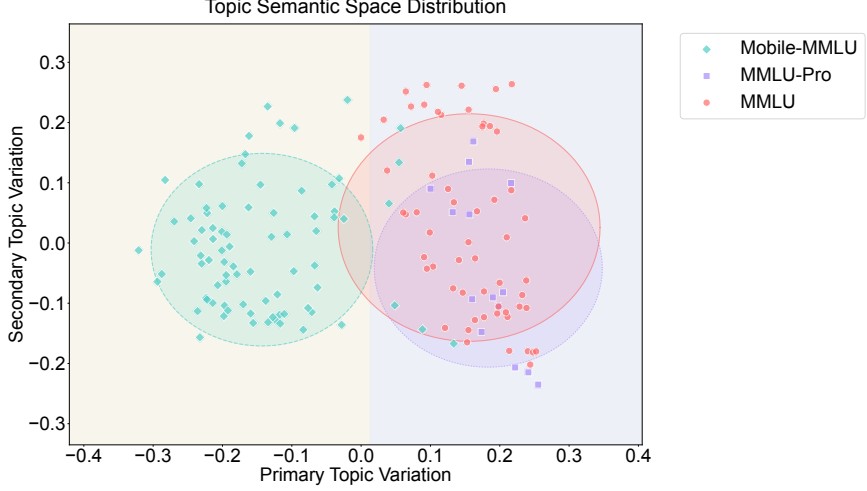

Figure 8: Topic distribution across Mobile-MMLU, MMLU, and MMLU-Pro benchmarks.

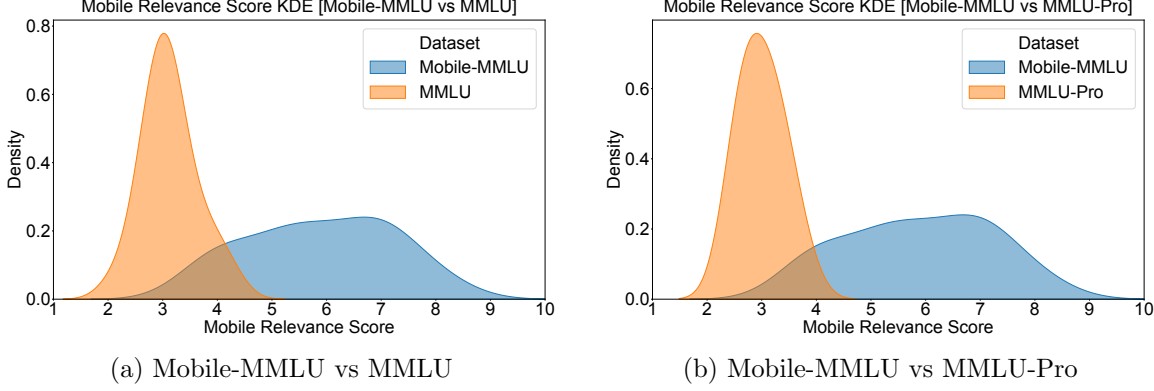

(a) Mobile-MMLU vs MMLU

(b) Mobile-MMLU vs MMLU-Pro

Figure 9: Distribution of MRScore comparing Mobile-MMLU against general benchmarks.

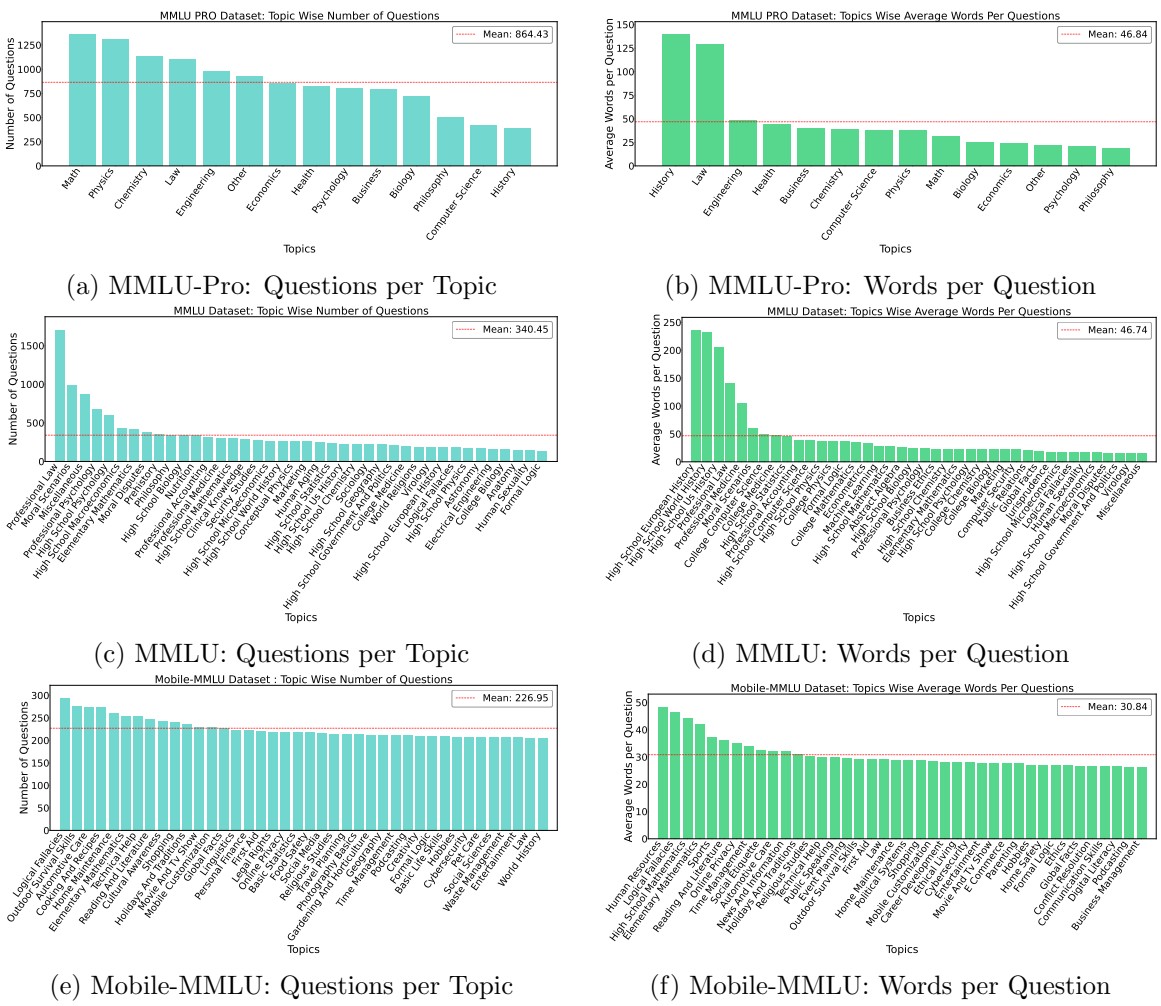

(a) MMLU-Pro: Questions per Topic

(b) MMLU-Pro: Words per Question

(c) MMLU: Questions per Topic

(d) MMLU: Words per Question

(e) Mobile-MMLU: Questions per Topic

(f) Mobile-MMLU: Words per Question

Figure 10: Distribution of questions and average words per question across topics in MMLU-Pro, MMLU, and Mobile-MMLU datasets (showing top 40 topics). The red dashed line indicates the mean value for each metric.

Our Mobile-MMLU consists of 80 topics, 16,186 questions, curated with the process discussed in the previous sections to evaluate small language models for the mobile specific use cases. Each topic includes multiple-choice questions designed to test both fundamental knowledge and real-world applications. Figure 10 shows the distribution of questions across top 40 topics for mobile-mmlu and mmlu dataset and all 14 topics for mmlu-pro, demonstrating the breadth and depth of our benchmark's coverage. Our questions focus on practical mobile usage scenarios and everyday tasks, ranging from "Cooking And Recipes" to "Digital Literacy" and "Travel-Planning", while also covering essential knowledge areas. Compared to traditional benchmarks like MMLU and MMLU-Pro, our questions are deliberately crafted to be more mobile-friendly, better reflecting real-world mobile interactions and information-seeking patterns.

Figure 10 also shows that our Mobile-MMLU and Mobile-MMLU-Pro differ significantly from both MMLU and MMLU-Pro. MMLU-Pro concentrates heavily on specialized academic fields with Math (1,350 questions), Physics (1,300 questions), and Chemistry (1,150 questions) dominating the distribution, and MMLU emphasizes traditional educational subjects like Professional Law (1,700 questions) and Moral Scenarios (1,000 questions), in contrast, Mobile-MMLU maintains a more balanced distribution with practical topics such as Logical Fallacies (290 questions), Survival Skills (275 questions), and Automotive Care (270 questions) that are more relevant to everyday mobile information needs. The question length analysis further highlights this distinction, Mobile-MMLU questions average around 30.84 words per question, with even the most detailed topics like Human Resources not exceeding 48 words, making them more suitable for mobile interfaces. This is in stark contrast to MMLU and MMLU-Pro, which average 46.74 and 46.84 words per question, respectively, with some topics like European History in MMLU containing questions averaging over 230 words in length.

## A.1 Comparison of Mobile-MMLU and Mobile-MMLU-Pro

Table 6 provides a detailed comparison between Mobile-MMLU and Mobile-MMLU-Pro, highlighting key differences in size and difficulty.

| Aspect | Mobile-MMLU | Mobile-MMLU-Pro |
|---|---|---|
| **Data Quality** | High Quality (General mobile-centric dataset) | High Quality (Refined dataset) |
| **Difficulty** | Easier for most models (Appropriate for general use) | More difficult for all models (Targets challenging queries) |
| **Question Types** | Broad range of question types (General mobile knowledge) | Focus on harder, more complex questions (Advanced difficulty) |
| **Size** | Larger in size (16,186) | Smaller in size (9,497) |
| **Scenario Used** | Suitable for general use (Wide applicability) | Ideal for users seeking small, challenging datasets (Consistent model performance) |

Table 6: Comparison of Mobile-MMLU and Mobile-MMLU-Pro datasets.

## A.2  Visualization of Model Size vs. Benchmark Performance

Figures 11 and 12 show the relationship between model parameter and performance across our standard and pro version of benchmarks. For the standard benchmark comparison in Figure 11, MMLU (blue dotted line) consistently shows higher performance than Mobile-MMLU (orange dotted line) across most model sizes, with scores 5∼10% higher for comparable models. Also, as we have mentioned in Section 5.2, model performance on MMLU is more concentrated and lacks distinction. In the pro version comparison of Figure 12, Mobile-MMLU-Pro scores are higher than MMLU-Pro for the same models, while model performance on MMLU-Pro is still more intensive than models on our Mobile-MMLU-Pro. Both benchmark families show a positive correlation between model size and performance, but with notably different absolute scores. Some smaller models like Qwen2.5-3B-Instruct achieve competitive performance on both benchmark variants, suggesting that architectural improvements can sometimes compensate for smaller parameter counts.

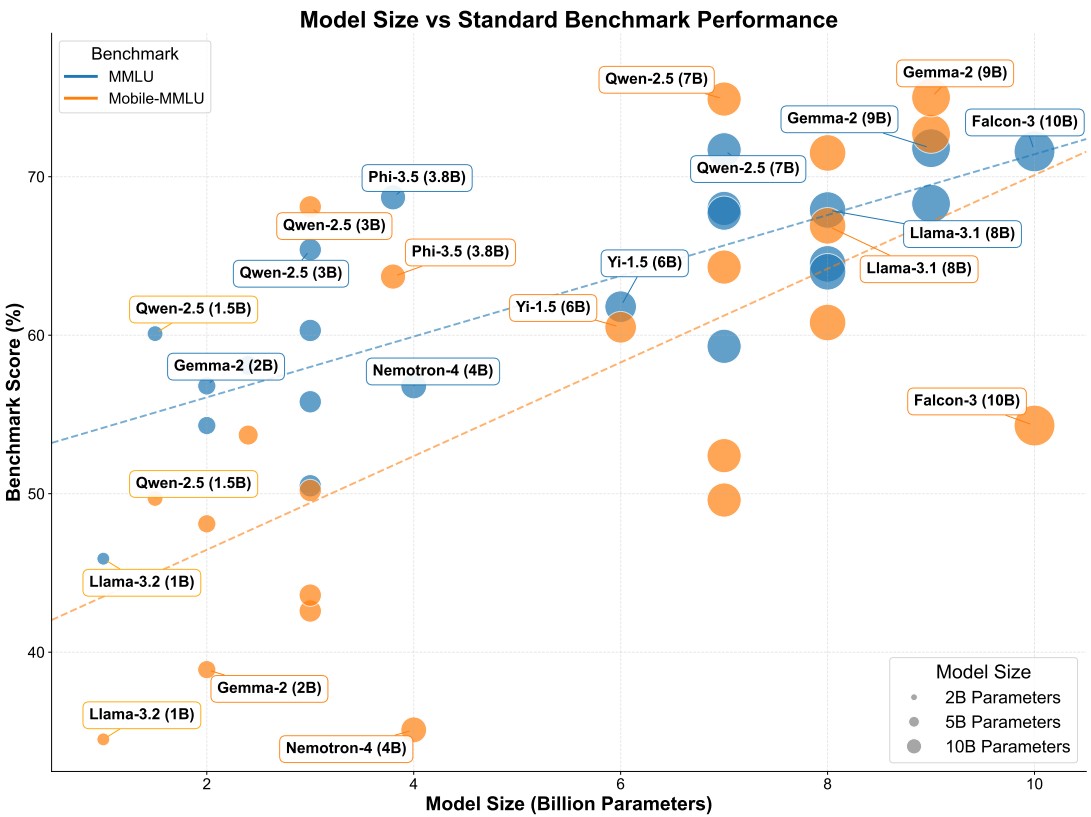

Figure 11: Model Size vs. Performance of Mobile-MMLU and MMLU.

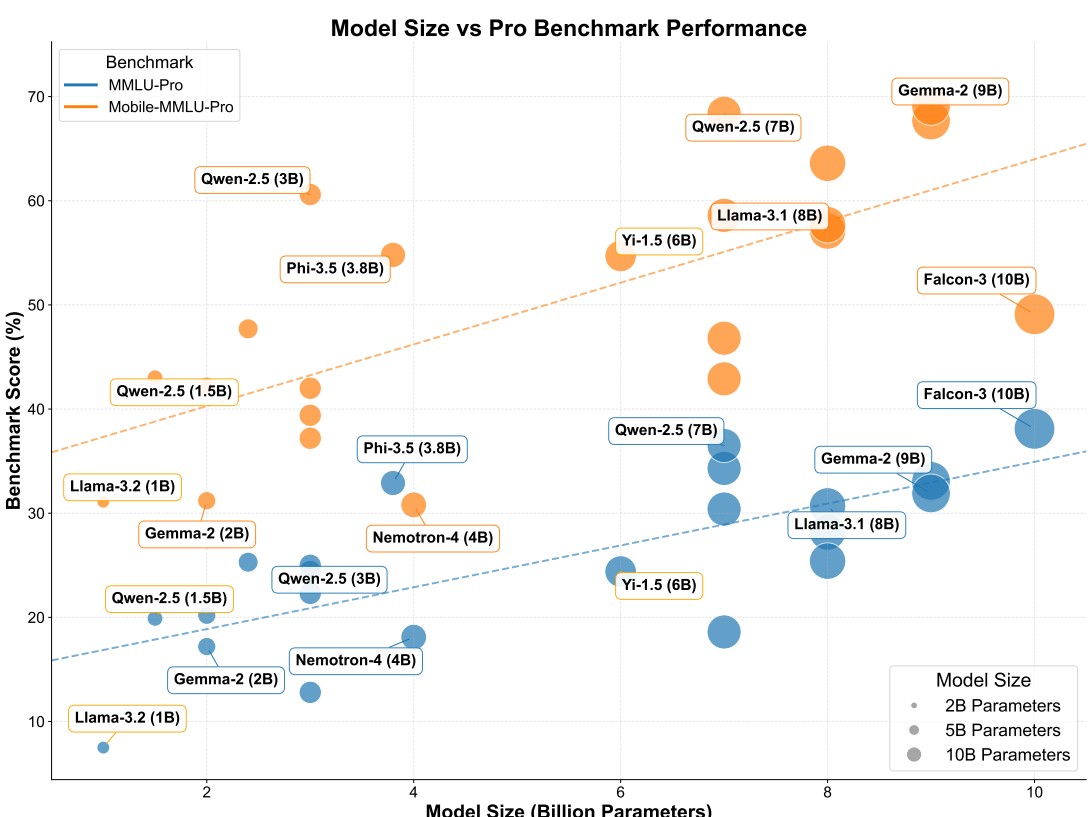

Figure 12: Model Size vs. Performance of Mobile-MMLU-Pro and MMLU-Pro.

## Appendix B. Examples from Our Mobile-MMLU Benchmark

We provide examples from our Mobile-MMLU benchmark to showcase the diversity of questions in our dataset. Each example consists of a question from one of the 80 fields covered, along with multiple-choice answer options and the correct answer. Tables 7, 8, and 9 present examples from three distinct fields: `Technical Help`, `Home Maintenance`, and `Ergonomics`. Additionally, Figures 5, 6, and 7 illustrate more examples from other fields.

---

**Field Name:** `Technical_help`
**Question:** `How can I connect my phone to a car's Bluetooth system?`
**Options:**
A. `To pair your phone with a car's Bluetooth system, start by switching on Bluetooth on your phone.  Next, enter the car's media system, find the Bluetooth options, and select "New Device" or "Join Device." Follow the detailed instructions on the screen to conclude the pairing procedure.`
B. `To synchronize your phone with a car's Bluetooth system, initially turn on Bluetooth on your phone.  Afterwards, head to the car's control panel, locate the connectivity settings, and pick "Discover Device" or "Link New Device." Follow the comprehensive prompts to finalize the connection process.`
C. `To connect your phone to a car's Bluetooth system, first enable Bluetooth on your phone.  Then, go to the car's infotainment system, navigate to the Bluetooth settings, and select "Add Device" or "Pair New Device." Follow the on-screen instructions to complete the pairing process.`
D. `To connect your phone to a car's Bluetooth system, first ensure Bluetooth is activated on your phone.  Then, access the car's infotainment system and proceed to the wireless settings menu.  Choose "Link Device" or "Connect New Device," and adhere to the displayed guidelines to finish the pairing process.`
**Correct Answer:** c

---

Table 7: Example Question from the `Technical Help` Field in the Mobile-MMLU Dataset.

## Appendix C. Top 40 Fields Different from MMLU

Here we present the top 40 fields from Mobile-MMLU that demonstrate the greatest differentiation from traditional MMLU benchmarks. These fields represent practical, mobile-centric knowledge domains that are highly relevant to daily smartphone usage but are absent or underrepresented in academic benchmarks like MMLU and MMLU-Pro.

The selection criteria prioritize fields with high Mobile-Relevance Scores that address real-world mobile use cases such as on-the-go problem solving, location-aware services, immediate information needs, and practical life skills. Unlike traditional MMLU topics that focus on academic subjects (e.g., Abstract Algebra, College Mathematics, Professional Law), these fields reflect the information-seeking behavior of mobile users who require actionable, context-sensitive knowledge.

Table 10 presents these 40 fields ranked by their Mobile-Relevance Scores, demonstrating the substantial gap between mobile user needs and traditional benchmark coverage. The highest-scoring fields (Travel Planning, First Aid, Photography & Smartphone Photography) achieve scores above 8.0, representing nearly three times the average relevance of

**Field Name:** `Home maintenance`
**Question:** What should I check if my washing machine is making loud banging noises during the spin cycle?(1) Ensure the washing machine is on a stable and flat surface, make sure the load is distributed evenly inside the drum, inspect the drum for any objects that might have been left behind, and check for worn or damaged drum bearings or suspension springs. (2) Check if the washing machine is balanced and level, ensure the load is evenly distributed, inspect the drum for foreign objects, and check for worn or damaged shock absorbers or suspension rods. (3) Check if the washing machine is correctly balanced and perfectly aligned, ensure the laundry load is properly distributed across the drum, inspect the inner drum for any foreign materials, and check for worn or damaged springs or suspension belts. (4) Verify that the washing machine is level and not tilted, confirm that the laundry is evenly spread within the drum, thoroughly inspect the drum for any foreign objects, and check for worn or damaged vibration dampers or suspension springs. Which of the statements given above are correct?
**Options:**
A. (2) and (4)
B. (1), (2) and (4)
C. (1) only
D. (2), (3) and (4)
**Correct Answer:** B

Table 8: Example Question from the `Home Maintenance` Field in the Mobile-MMLU Dataset.

**Field Name:** `Ergonomics`
**Question:** What is the best way to hold a smartphone to reduce strain?
**Options:**
A. Hold the smartphone at waist level, which causes significant neck strain, with a relaxed grip to reduce strain while maintaining a comfortable posture .
B. Hold the smartphone slightly above eye level, which is uncomfortable, with a firm grip.
C. Hold the smartphone at chest level, which still requires neck tilting, with a relaxed grip to minimize strain.
D. Hold the smartphone at eye level with a relaxed grip to reduce strain.
**Correct Answer:** D

Table 9: Example Question from the Ergonomics Field in the Mobile-MMLU Dataset.

standard MMLU topics (3.13). This stark contrast validates our hypothesis that existing benchmarks inadequately capture the knowledge requirements for mobile language model applications.

| Rank | Field | Mobile-Relevance Score |
|------|-------|------------------------|
| 1 | Travel Planning | 8.43 |
| 2 | First Aid | 8.41 |
| 3 | Photography & Smartphone Photography | 8.05 |
| 4 | Shopping | 7.96 |
| 5 | Technical Help | 7.88 |
| 6 | Digital Literacy | 7.79 |
| 7 | Mobile Customization | 7.74 |
| 8 | Cybersecurity | 7.65 |
| 9 | Online Privacy | 7.43 |
| 10 | Social Media | 7.38 |
| 11 | Automotive Care | 7.33 |
| 12 | Personal Finance | 7.32 |
| 13 | Outdoor Survival Skills | 7.26 |
| 14 | Mental Health | 7.15 |
| 15 | Cooking & Recipes | 7.11 |
| 16 | Sports | 7.11 |
| 17 | Home Safety | 7.06 |
| 18 | Time Management | 7.05 |
| 19 | Fashion & Style | 7.00 |
| 20 | Food Safety | 7.00 |
| 21 | Physical Fitness | 7.00 |
| 22 | Nutrition & Diet | 6.93 |
| 23 | Pet Care | 6.85 |
| 24 | Creativity | 6.81 |
| 25 | Basic Life Skills | 6.80 |
| 26 | Social Etiquette | 6.80 |
| 27 | Parenting | 6.79 |
| 28 | Medical & Health Knowledge | 6.78 |
| 29 | Home Maintenance | 6.73 |
| 30 | Entertainment | 6.70 |
| 31 | Weather Forecasting | 6.50 |
| 32 | Digital Detox | 6.37 |
| 33 | Relationships | 6.24 |
| 34 | E-commerce | 6.22 |
| 35 | News & Information | 6.15 |
| 36 | Legal Rights | 6.14 |
| 37 | Teens & Youth | 6.13 |
| 38 | Environmental & Sustainable Living | 6.12 |

Table 10 Continued from previous page

| Rank | Field | Mobile-Relevance Score |
|------|-------|------------------------|
| 39 | Public Speaking | 6.00 |
| 40 | Waste Management | 6.00 |

Table 10: Top 40 Mobile-Specific Fields Different from Traditional MMLU Benchmarks.

## C.1 Key Differentiating Characteristics

The fields listed in Table 10 exhibit several key characteristics that distinguish them from traditional MMLU domains. These fields demonstrate exceptional practical applicability, as they address immediate, real-world problems that mobile users encounter in their daily lives, ranging from travel planning and emergency first aid to technical troubleshooting and personal finance management. Unlike the theoretical and academic focus of traditional MMLU subjects, these domains provide actionable knowledge that users can immediately apply to solve concrete problems.

## Appendix D. Data Hierarchy

Table 11 shows the complete hierarchy of our Mobile-MMLU dataset. It covers most of the common knowledge domains relevant to mobile users' daily information needs. The dataset is organized into 9 major categories: `Academic & Learning, Business & Career, Technology & Digital, Health & Safety, Lifestyle & Personal, Home & Family, Culture & Society, and Environment, with an additional category of Miscellaneous`. These categories are further divided into 27 subcategories and 80 distinct topics, with a total population of 16,186 question-answer pairs. The `Lifestyle & Personal` category contains the highest number of topics (17) covering various aspects of personal growth and daily interests. The `Academic & Learning` category follows with 15 topics that address fundamental educational areas from elementary mathematics to world history. This hierarchical structure ensures comprehensive coverage of the knowledge domains that mobile users frequently encounter and query, bridging the gap in the existing general purpose benchmarks.

Table 11: Hierarchical Structure of Categories, Subcategories, and Topics with Population of Mobile-MMLU.

| Category | Subcategory | Topic | Population |
|----------|-------------|-------|------------|
| Academic & Learning | Basic Mathematics | Elementary Mathematics | 254 |
| | | High School Mathematics | 200 |
| | | Basic Statistics | 219 |
| | Basic Sciences | Conceptual Physics | 194 |
| | | Science Fundamentals | 191 |
| | Critical Thinking | Formal Logic | 210 |
| | | Logical Fallacies | 293 |
| | Education | Education Techniques | 146 |
| | | Reading & Literature | 248 |
| | | Writing Skills | 203 |

Table 11 continued from the previous page

| Category | Subcategory | Topic | Population |
|---|---|---|---|
| | Social Sciences | Linguistics | 223 |
| | | Social Sciences | 207 |
| | | Political Systems | 193 |
| | | World History | 206 |
| | | Geography | 211 |
| Business & Career | Business Studies | Project Management | 176 |
| | | Human Resources | 144 |
| | | Business Management | 167 |
| | | Marketing & Sales Strategies | 162 |
| | Personal Business | Personal Finance | 223 |
| | | E Commerce | 186 |
| | | Shopping | 241 |
| | | Accounting | 205 |
| | Communication | Communication Skills | 134 |
| | | Social Etiquette | 200 |
| | | Public Speaking | 158 |
| Technology & Digital | Digital Literacy | Digital Literacy | 198 |
| | | Technical Help | 254 |
| | | Mobile Customization | 230 |
| | Privacy & Security | Cybersecurity | 208 |
| | | Online Privacy | 219 |
| | Social Media | Social Media | 217 |
| | | Digital Detox | 196 |
| Health & Safety | Health & Wellness | Mental Health | 130 |
| | | Physical Fitness | 190 |
| | | Medical & Health Knowledge | 183 |
| | | Ergonomics | 204 |
| | Everyday Safety | First Aid | 221 |
| | | Outdoor Survival Skills | 277 |
| | | Automotive Care | 275 |
| Lifestyle & Personal | Daily Life Skills | Basic Life Skills | 209 |
| | | Time Management | 211 |
| | | Conflict Resolution | 152 |
| | | Event Planning | 201 |
| | Personal Growth | Creativity | 210 |
| | | Emotional Intelligence | 133 |
| | | Personal Branding | 186 |
| | | Career Development | 166 |
| | Lifestyle | Fashion & Style | 200 |
| | | Travel Planning | 214 |
| | | Sports | 188 |
| | | Gardening & Horticulture | 212 |
| | Entertainment | Entertainment | 207 |
| | | Movie & TV Show | 230 |
| | | Podcasting | 211 |
| | | Hobbies | 208 |
| | | Photography Basics | 214 |
| Home & Family | Home & Living | Home Safety | 189 |
| | | Pet Care | 208 |

Table 11 continued from the previous page

| Category | Subcategory | Topic | Population |
|----------|-------------|-------|-----------|
| | Family | Waste Management | 207 |
| | | Home Maintenance | 261 |
| | | Parenting | 144 |
| | | Relationships | 173 |
| | | Teens & Youth | 161 |
| | Food & Cooking | Cooking & Recipes | 274 |
| | | Food Safety | 219 |
| | | Nutrition & Diet | 151 |
| Culture & Society | Arts & Design | Art Techniques & Architecture | 175 |
| | | Interior Design | 200 |
| | Culture & Religion | Cultural Awareness | 243 |
| | | Religious Studies | 215 |
| | | Holidays & Traditions | 236 |
| | Legal | Legal Rights | 219 |
| | | Law | 206 |
| | Ethics & Morality | Ethical Living | 171 |
| | | Ethics | 172 |
| Environment | Environment | Sustainable Living | 178 |
| | Weather | Weather Forecasting | 205 |
| Miscellaneous | Miscellaneous | Global Facts | 228 |
| | | News & Information | 203 |

## Appendix E. Mobile Relevance Score

We calculate the Mobile-Relevance Score for all topics across different benchmarks using Equation 3 as discussed in Section A. Our analysis reveals that Mobile-MMLU achieves the highest average relevance score of 5.88, almost twice as high as standard MMLU (3.13) and MMLU-Pro (3.00). This significant difference is further illustrated in Table 12, where Mobile-MMLU contains 21 highly relevant topics (scoring $\geq 7.0$) and 38 moderately relevant topics (scoring 5.0-6.9), while neither MMLU nor MMLU-Pro include any topics in these higher relevance categories.

The detailed breakdown in Table 13 demonstrates that the highest scoring topics in Mobile-MMLU are directly related to daily mobile use cases, with Travel Planning (8.43), First Aid (8.41), and Photography & Smartphone Photography (8.05) leading the rankings. This distribution highlights how traditional benchmarks fail to adequately capture the information needs of mobile phone users, substantially reducing the practical utility of smaller language models designed for mobile applications. Our dataset successfully addresses these limitations by focusing on knowledge domains that are directly applicable to mobile users' information-seeking behavior.

Table 12: Statistical Overview of Relevance Scores Across MMLU Variants.

| Benchmark | Avg. Score | Topics $\geq$7.0 | Topics 5.0-6.9 | Topics 3.0-4.9 | Topics <3.0 |
|-----------|-----------|---------|-------------|-------------|-----------|
| **Mobile MMLU** | **5.88** | **21** | 38 | 21 | 0 |
| **Mobile MMLU-Pro** | **5.97** | **23** | **36** | 21 | 0 |
| MMLU | 3.13 | 0 | 0 | **29** | 18 |
| MMLU-Pro | 3.00 | 0 | 0 | 6 | **8** |

Table 13: Complete Relevance Scores Grouped by Relevance Range.

| Topic | Benchmark | Relevance Score |
|---|---|---|
| **High Relevance (7.0-10.0)** | | |
| Travel Planning | Mobile MMLU | 8.43 |
| First Aid | Mobile MMLU | 8.41 |
| Photography & Smartphone Photography | Mobile MMLU | 8.05 |
| Shopping | Mobile MMLU | 7.96 |
| Technical Help | Mobile MMLU | 7.88 |
| Digital Literacy | Mobile MMLU | 7.79 |
| Mobile Customization | Mobile MMLU | 7.74 |
| Cybersecurity | Mobile MMLU | 7.65 |
| Online Privacy | Mobile MMLU | 7.43 |
| Social Media | Mobile MMLU | 7.38 |
| Automotive Care | Mobile MMLU | 7.33 |
| Personal Finance | Mobile MMLU | 7.32 |
| Outdoor Survival Skills | Mobile MMLU | 7.26 |
| Mental Health | Mobile MMLU | 7.15 |
| Cooking & Recipes | Mobile MMLU | 7.11 |
| Sports | Mobile MMLU | 7.11 |
| Time Management | Mobile MMLU | 7.05 |
| Home Safety | Mobile MMLU | 7.06 |
| Fashion & Style | Mobile MMLU | 7.00 |
| Food Safety | Mobile MMLU | 7.00 |
| Physical Fitness | Mobile MMLU | 7.00 |
| **Medium Relevance (5.0-6.9)** | | |
| Nutrition & Diet | Mobile MMLU | 6.93 |
| Pet Care | Mobile MMLU | 6.85 |
| Creativity | Mobile MMLU | 6.81 |
| Basic Life Skills | Mobile MMLU | 6.80 |
| Social Etiquette | Mobile MMLU | 6.80 |
| Parenting | Mobile MMLU | 6.79 |
| Medical & Health Knowledge | Mobile MMLU | 6.78 |
| Home Maintenance | Mobile MMLU | 6.73 |
| Entertainment | Mobile MMLU | 6.70 |
| Weather Forecasting | Mobile MMLU | 6.50 |
| Digital Detox | Mobile MMLU | 6.37 |
| Relationships | Mobile MMLU | 6.24 |
| E-commerce | Mobile MMLU | 6.22 |
| News & Information | Mobile MMLU | 6.15 |
| Legal Rights | Mobile MMLU | 6.14 |
| Teens & Youth | Mobile MMLU | 6.13 |
| Environmental & Sustainable Living | Mobile MMLU | 6.12 |
| Public Speaking | Mobile MMLU | 6.00 |
| Waste Management | Mobile MMLU | 6.00 |
| Gardening & Horticulture | Mobile MMLU | 5.95 |
| Interior Design | Mobile MMLU | 5.80 |
| Geography | Mobile MMLU | 5.76 |
| Emotional Intelligence | Mobile MMLU | 5.69 |
| Marketing & Sales Strategies | Mobile MMLU | 5.69 |
| Event Planning | Mobile MMLU | 5.65 |
| Ergonomics | Mobile MMLU | 5.60 |

Table 13 – Continued from previous page

| Topic | Benchmark | Relevance Score |
|---|---|---|
| Hobbies | Mobile MMLU | 5.60 |
| Career Development | Mobile MMLU | 5.38 |
| Global Facts | Mobile MMLU | 5.32 |
| Holidays & Traditions | Mobile MMLU | 5.30 |
| Cultural Awareness | Mobile MMLU | 5.29 |
| Personal Branding | Mobile MMLU | 5.22 |
| Movie & TV Shows | Mobile MMLU | 5.22 |
| Podcasting | Mobile MMLU | 5.19 |
| Communication & Public Speaking | Mobile MMLU | 5.15 |
| Business Management | Mobile MMLU | 5.13 |
| Project Management | Mobile MMLU | 5.12 |
| Education Techniques | Mobile MMLU | 5.07 |

*(Note: All medium relevance topics are from Mobile MMLU)*

| **Low-Medium Relevance (3.0-4.9)** | | |
|---|---|---|
| Law | Mobile MMLU | 4.95 |
| Conflict Resolution | Mobile MMLU | 4.87 |
| Elementary Mathematics | Mobile MMLU | 4.80 |
| Writing Skills | Mobile MMLU | 4.70 |
| Religious Studies | Mobile MMLU | 4.38 |
| Conceptual Physics | Mobile MMLU | 4.37 |
| Science Fundamentals | Mobile MMLU | 4.32 |
| Ethical Living | Mobile MMLU | 4.29 |
| Accounting | Mobile MMLU | 4.20 |
| High School Mathematics | Mobile MMLU | 4.20 |
| Basic Statistics | Mobile MMLU | 4.19 |
| Social Sciences | Mobile MMLU | 4.10 |
| Ethics | Mobile MMLU | 4.06 |
| Human Resources | Mobile MMLU | 4.00 |
| Art Techniques & Architecture | Mobile MMLU | 3.88 |
| Linguistics | Mobile MMLU | 3.82 |
| Political Systems | Mobile MMLU | 3.74 |
| Reading & Literature | Mobile MMLU | 3.71 |
| Logical Fallacies | Mobile MMLU | 3.66 |
| Formal Logic | Mobile MMLU | 3.52 |
| World History | Mobile MMLU | 3.50 |
| Miscellaneous | MMLU | 4.34 |
| Elementary Mathematics | MMLU | 4.20 |
| Human Sexuality | MMLU | 4.14 |
| Computer Security | MMLU | 4.00 |
| Virology | MMLU | 3.94 |
| Clinical Knowledge | MMLU | 3.93 |
| Nutrition | MMLU | 3.73 |
| World Religions | MMLU | 3.68 |
| Human Aging | MMLU | 3.67 |
| Management | MMLU | 3.64 |
| Marketing | MMLU | 3.60 |
| Public Relations | MMLU | 3.42 |
| High School Computer Science | MMLU | 3.40 |
| High School Geography | MMLU | 3.36 |
| High School Government & Politics | MMLU | 3.33 |

*Continued on next page*

Table 13 – Continued from previous page

| Topic | Benchmark | Relevance Score |
|---|---|---|
| Security Studies | MMLU | 3.33 |
| Global Facts | MMLU | 3.27 |
| US Foreign Policy | MMLU | 3.27 |
| Medical Genetics | MMLU | 3.27 |
| High School Macroeconomics | MMLU | 3.26 |
| Electrical Engineering | MMLU | 3.25 |
| Anatomy | MMLU | 3.14 |
| High School Biology | MMLU | 3.12 |
| High School Microeconomics | MMLU | 3.12 |
| College Medicine | MMLU | 3.11 |
| High School Psychology | MMLU | 3.10 |
| Professional Psychology | MMLU | 3.09 |
| Sociology | MMLU | 3.09 |
| Conceptual Physics | MMLU | 3.08 |
| International Law | MMLU | 3.08 |
| Astronomy | MMLU | 3.06 |
| High School Statistics | MMLU | 3.04 |
| Professional Accounting | MMLU | 3.03 |
| Professional Medicine | MMLU | 3.00 |
| Business Ethics | MMLU | 3.00 |
| Business | MMLU-Pro | 3.73 |
| Other | MMLU-Pro | 3.55 |
| Psychology | MMLU-Pro | 3.31 |
| Health | MMLU-Pro | 3.30 |
| Biology | MMLU-Pro | 3.19 |
| Economics | MMLU-Pro | 3.19 |
| **Low Relevance (<3.0)** | | |
| High School Chemistry | MMLU | 2.95 |
| High School Mathematics | MMLU | 2.93 |
| Machine Learning | MMLU | 2.92 |
| College Physics | MMLU | 2.91 |
| Moral Scenarios | MMLU | 2.89 |
| Prehistory | MMLU | 2.86 |
| Econometrics | MMLU | 2.83 |
| High School Physics | MMLU | 2.81 |
| Formal Logic | MMLU | 2.79 |
| Logical Fallacies | MMLU | 2.78 |
| Philosophy | MMLU | 2.76 |
| College Biology | MMLU | 2.75 |
| Jurisprudence | MMLU | 2.73 |
| College Computer Science | MMLU | 2.73 |
| Moral Disputes | MMLU | 2.68 |
| College Chemistry | MMLU | 2.60 |
| College Mathematics | MMLU | 2.55 |
| Abstract Algebra | MMLU | 2.55 |
| Professional Law | MMLU | 2.49 |
| High School US History | MMLU | 2.45 |
| High School European History | MMLU | 2.11 |
| High School World History | MMLU | 2.08 |
| Computer Science | MMLU-Pro | 2.95 |

*Continued on next page*

Table 13 – Continued from previous page

| Topic | Benchmark | Relevance Score |
|---|---|---|
| Mathematics | MMLU-Pro | 2.91 |
| Physics | MMLU-Pro | 2.84 |
| Philosophy | MMLU-Pro | 2.80 |
| Chemistry | MMLU-Pro | 2.63 |
| Engineering | MMLU-Pro | 2.59 |
| History | MMLU-Pro | 2.55 |
| Law | MMLU-Pro | 2.45 |

# Appendix F. Prompts

## F.1 Prompt Template for Question Generation

---

**Prompt Template for Question Generation**

Act as a professional benchmark creator. Your task is to generate 200 questions for the given field. The questions should be comprehensive, covering all major aspects of the field, and should span a range of difficulty levels: easy, medium, and hard.

## Instruction ##
Generate 200 questions related to the field of {field}, which is {explanation}.
Output each question on a new line.

## Hint ##
These questions will be used to create a benchmark for evaluating large language models (LLMs) in mobile use cases. Ensure that the questions are realistic, practical, and reflect the types of queries mobile users might ask in their daily lives.

## Example Question Starters ##
1. How can I...
2. When should I...
3. How do I...
4. What is...

---

---

**Prompt Template for Generating Challenging Questions**

You are a professional benchmark question creator. Your task is to generate 120 non-redundant, highly challenging questions for evaluating large language models (LLMs). The questions should test the models' abilities in commonsense reasoning, deep problem-solving, and their applicability to real-world, everyday scenarios.

## Instruction ##
Generate exactly 120 unique, highly challenging, and complex questions in the field of {field}. These questions should represent difficult, yet realistic queries that a mobile phone user might ask, blending commonsense reasoning, deep analysis, and advanced problem-solving. Focus on creating complex scenarios that push the model's capabilities to the limit in handling intricate tasks and challenges.

## Guidelines##

- You must generate exactly 120 questions—no more, no less. You will be penalized if you fail to meet this requirement.
- All questions should be complex, requiring the model to deeply reason, analyze, or problem-solve across multiple contexts or data points.
- Each question must be uniquely difficult, without any redundancy or repetition, ensuring it tests the model's understanding thoroughly.
- Ensure that the questions reflect real-world daily use cases and are phrased in a style that users would naturally ask on mobile devices.
- You may vary the length of questions. Diversity in phrasing and structure is encouraged.

## Output Format ##
- Do not number the questions.
- Present each question on a new line for clarity.

---

## F.2 Prompt for Generating Mobile Relevance Score

---

**Mobile Relevance Evaluation Prompt**

You are an expert in evaluating the relevance of different types of questions for mobile devices. Assess a given question using the criteria of practical value, mobile-friendliness, and usage patterns to return a relevance score from 1 to 10.

- - - - - - - - - - - - - - - - - - - - - - - - - - - - - - - - - - - - - - - - - - - - - - - - - - - - - -

**Practical Value:** Is the information needed in daily life or in real-world tasks? Does it solve problems or support decisions while away from a desk?

**Mobile-Friendliness:** Can the content be effectively consumed on mobile in terms of its format and interactivity?

**Usage Patterns:** Is this the kind of information someone would typically look up on their phone? Is it useful for in-the-moment decision-making?

- - - - - - - - - - - - - - - - - - - - - - - - - - - - - - - - - - - - - - - - - - - - - - - - - - - - - -

**Question**:
{question}

- - - - - - - - - - - - - - - - - - - - - - - - - - - - - - - - - - - - - - - - - - - - - - - - - - - - - -

Return **only the numeric score** from 1 to 10 without any explanation.

---

## F.3 Prompt for Generating Ground Truth Answer

**Instructions for Generating a Ground Truth Answer**

You are an expert in answering questions across a diverse range of fields. Your task is to generate a correct and **concise** ground truth answer of reasonable length for the given question. Do not add a lot of details and **never** use point-wise answers, since this ground truth will be one of the options for multiple-choice questions.

**Instructions**:

1. Generate the correct and **concise** ground truth answer for the given question.

2. Include relevant context when necessary and base the answer on well-established facts.

3. Use clear, precise, and concise language and avoid ambiguity.

4. Provide a confidence score for your answer, which is between 1 to 5.

   - 5: Very confident
   - 1: Very uncertain

5. Provide a 'yes' tag if the question can be rewritten without multiple-choice options, and 'no' if it cannot.

**Given Question**:
{question}

**Output format**:

```
Ground_Truth:
Confidence_Score:
Tag:
```

### F.4 Prompt for Generating Incorrect Answers

---

**Instructions for Generating Incorrect Answers**

You are an expert at generating **incorrect answers**. For each question provided along with the ground truth answer, your task is to generate **three** incorrect answers that closely resemble the ground truth answer. Modify **only key terms**—specifically important ones—in the correct answer to create incorrect answers that appear plausible but are distinct. Ensure that the length of the incorrect answers is longer than the ground truth answer, and **never shorter**. Do not make the incorrect answers too obvious or very easy to distinguish from the correct one, and also ensure they are **relevant** and not just paraphrasing of the correct answers.

- - - - - - - - - - - - - - - - - - - - - - - - - - - - - - - - - - - - - - - - - - - - - - - - -

**Instructions**:

1. Modify only key terms in the correct answer to create each incorrect answer.

2. Ensure that the length of the incorrect answers is longer than the ground truth answer, and **never shorter**. You may add more detailed explanations to the unmodified parts of the ground truth to increase the length.

3. Make each incorrect answer distinct, modifying different aspects of the correct answer.

4. Avoid using revealing words like 'mainly', 'basically', 'only', etc., which might make incorrect answers obvious.

5. Do not simply negate the correct answer and never paraphrase with synonyms. Simple paraphrasing with synonyms will never make the statement incorrect.

6. Consider the context of the question and ground truth when generating incorrect answers.

7. Ensure modified terms are within the same domain or category as the original terms.

8. The incorrect answers should be plausible and logical, but still incorrect.

9. Each output should start on a new line.

- - - - - - - - - - - - - - - - - - - - - - - - - - - - - - - - - - - - - - - - - - - - - - - - -

**Question**: {question}
**Ground Truth Correct Answer**: {ground_truth_answer}

- - - - - - - - - - - - - - - - - - - - - - - - - - - - - - - - - - - - - - - - - - - - - - - - -

**Output format**:

```
Incorrect_Answer_A:
Incorrect_Answer_B:
Incorrect_Answer_C:
```

---

## Appendix G. Topic and Category Wise Performance Variations Models

Figure 18 illustrates the comparative performance of seven language models across 80 topics in our Mobile-MMLU dataset. The heatmap reveals significant performance variations both in models and topics, highlighting the critical need for mobile-specific evaluation benchmarks. Traditional benchmarks such as MMLU and MMLU-Pro fail to adequately capture the diverse knowledge domains essential for effective mobile applications, as demonstrated by their substantially lower Mobile-Relevance Scores shown in Section A. Our topic-specific analysis provides valuable insight for selecting appropriate models for different mobile application scenarios based on their relative strengths in domains most relevant to specific use cases.

Figure 13 presents a radar chart visualizing model performance in nine distinct knowledge categories critical for mobile applications. We observe that larger models generally outperform their smaller counterparts within the same family (Qwen2.5-7B vs. 3B, Gemma-2-9b vs. 2b), with Gemma-2-9b demonstrating particularly strong performance in Technology & Digital. In contrast, the Academic & Learning category exhibits greater variability between models, suggesting that traditional academic evaluations may not di-

Figure 13: Radar chart showing the comparative accuracy of seven language models across nine knowledge categories relevant to mobile users.

rectly translate to mobile-relevant performance. These patterns confirm that mobile-specific benchmarking is essential for developing models that serve mobile users effectively.

Figures 14 and 15 provide additional information by identifying the easiest and most challenging topics in the Mobile-MMLU data set. Science Fundamentals achieves the highest mean accuracy (0.690) in all models, followed closely by Conceptual Physics (0.686) and Mental Health (0.684). In particular, many high-relevance mobile topics such as Digital Detox (0.681), Nutrition & Diet (0.681), and Project Management (0.680) appear among the best performing tasks, indicating promising capabilities for practical mobile applications. In contrast, Figure 15 highlights the most challenging topics, with Elementary Mathematics showing surprisingly low performance (0.479 mean accuracy). More concerning for mobile applications are the difficulties models face with highly mobile-relevant domains: Travel Planning (0.605), Technical Help (0.602), and Shopping (0.583). These are precisely the practical, everyday tasks that mobile users frequently need assistance with, yet they pose significant challenges for current models. Such findings would remain hidden when using conventional benchmarks like MMLU and MMLU-Pro, which lack sufficient coverage of these mobile-critical domains. These results validate the contribution of Mobile-MMLU as an essential evaluation framework to develop more capable and useful language models for mobile devices, addressing the practical needs that traditional benchmarks overlook.

## Appendix H. Answer Order Sensitivity

Figures 16 and 17 illustrate how model performance varies with different answer orderings. Our analysis reveals two significant patterns in this sensitivity testing. First, when examining the randomized variants (R1-R4), we observe minimal performance fluctuations across all models (typically within ±3%), indicating

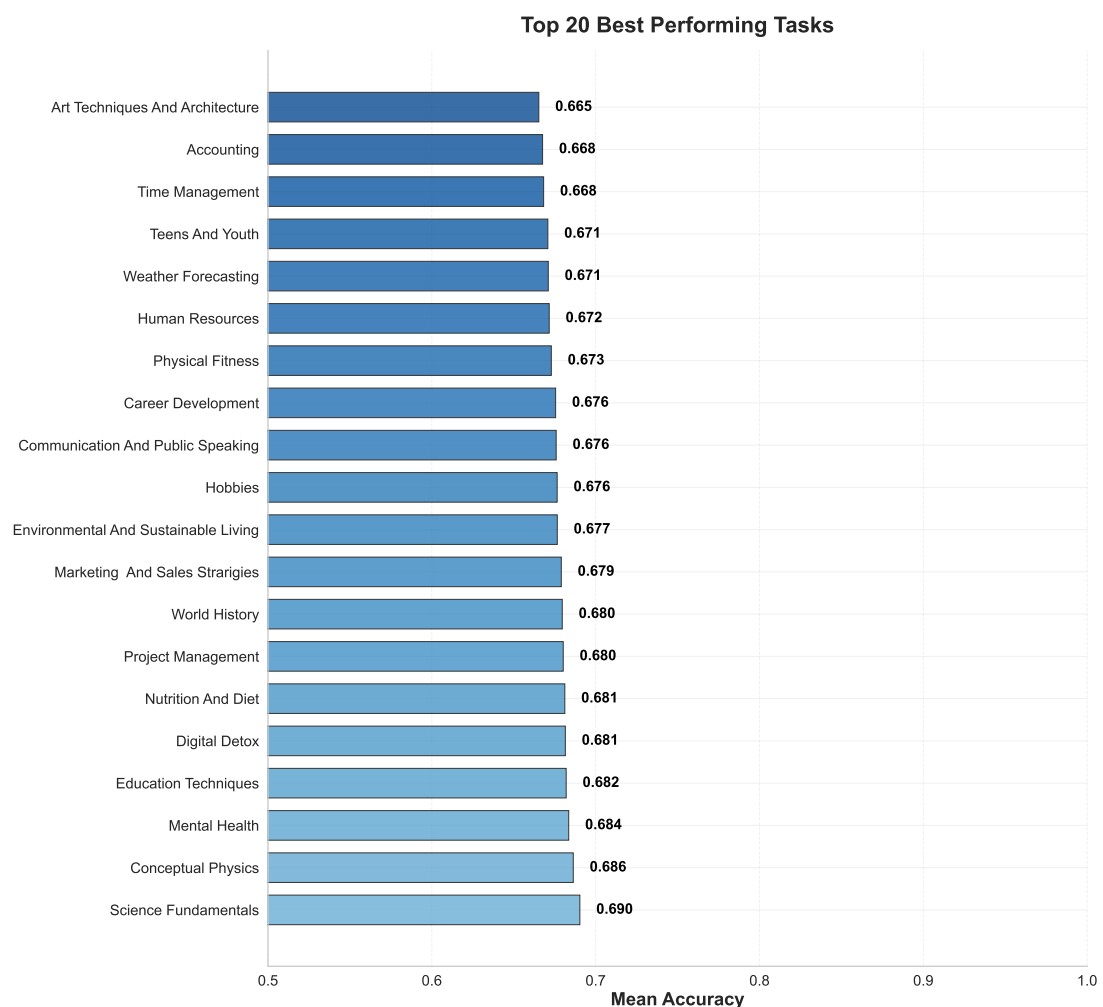

Figure 14: Top 20 best-performing topics across models in the Mobile-MMLU dataset, highlighting domains where current models show promise for mobile applications.

that the benchmark maintains consistency when answer positions are randomized. This stability suggests that Mobile-MMLU provides reliable and robust evaluation when using randomized answer positions.

Second, we find substantial performance variations when correct answers are fixed in specific positions (A, B, C, D). For instance, Gemma-2-2b shows a 28.77% improvement when correct answers appear in position A, while Phi-3.5-mini demonstrates a 23.96% improvement with position D. Conversely, several models show performance degradation with certain positions—notably Qwen2.5-7B (-20.08%) and Phi-3.5-mini (-17.93%) with position A. These pronounced variations reveal positional biases inherent in different model architectures.

The consistent performance across randomized variants compared to high variation with fixed positions demonstrates that position bias significantly impacts model evaluation. This finding has important implications for benchmark design, suggesting that fixed-position evaluations may artificially inflate or depress model scores based on their inherent positional preferences rather than their actual knowledge capabilities. Stable performance with randomized answer orders confirms that Mobile-MMLU provides a more objective and robust evaluation.

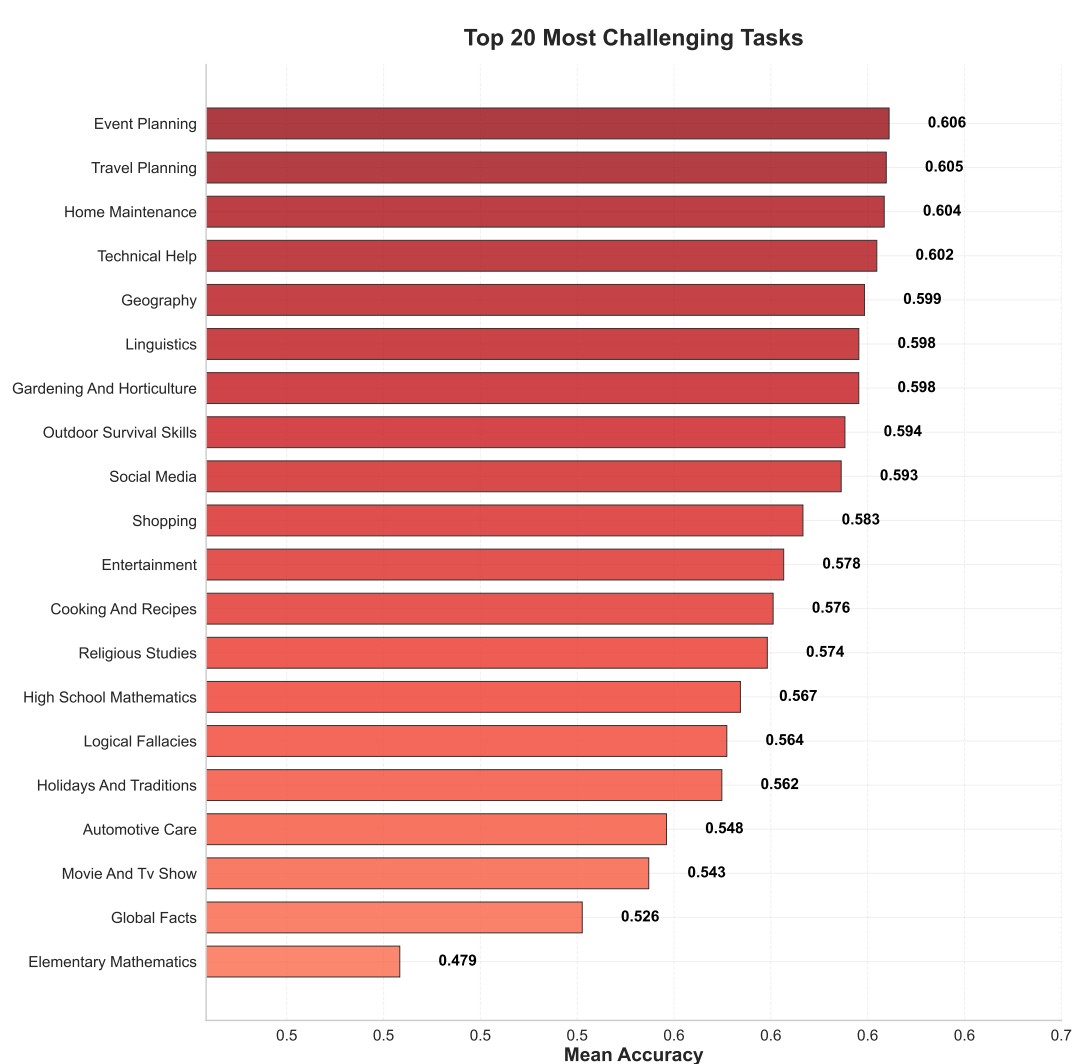

Figure 15: Top 20 most challenging topics in the Mobile-MMLU dataset, identifying critical areas for improvement in mobile-oriented language models.

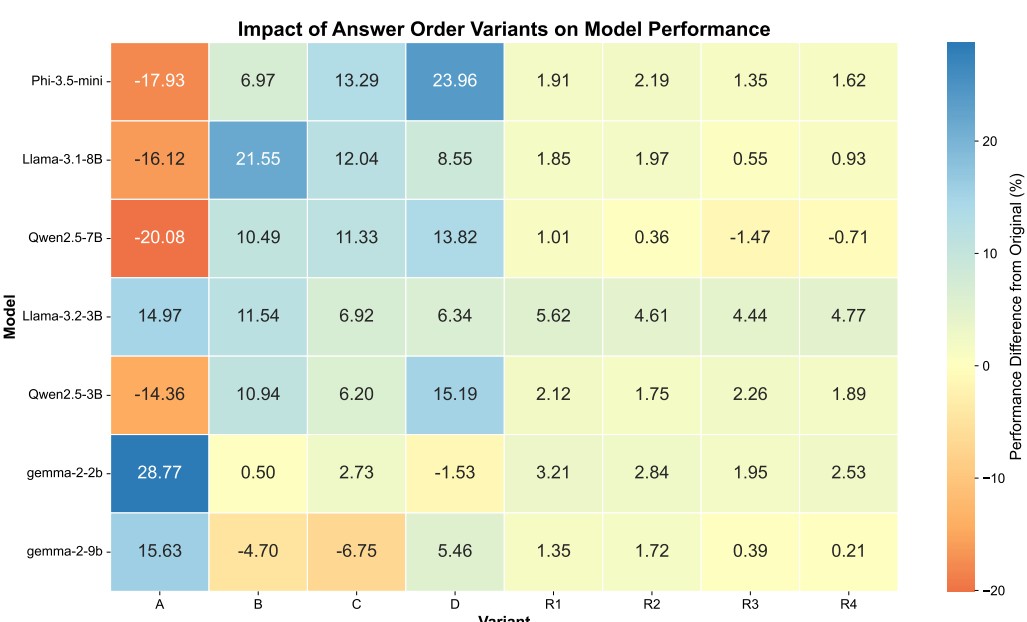

Figure 16: Impact of answer order variant in the model performance. Heatmap shows the performance difference from original model in terms of percentage. Note that, Variations because of different options shows the option led position bias in different model. However we also observe that in case of random options, all variants have almost similar performance which shows the robustness of our benchmark.

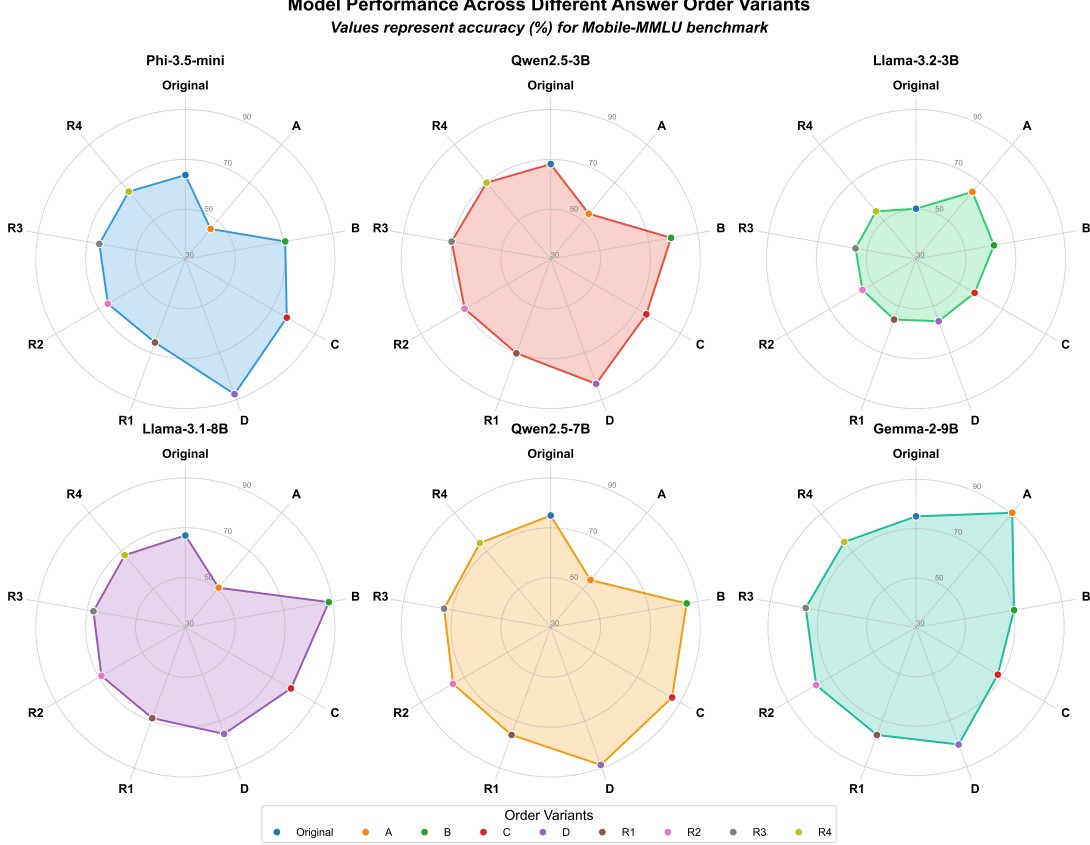

Figure 17: Answer Order Sensitivity Across Different Language Models: Impact of answer order variant in the model performance. Radar plot shows the performance for different options, different random orders and different models.

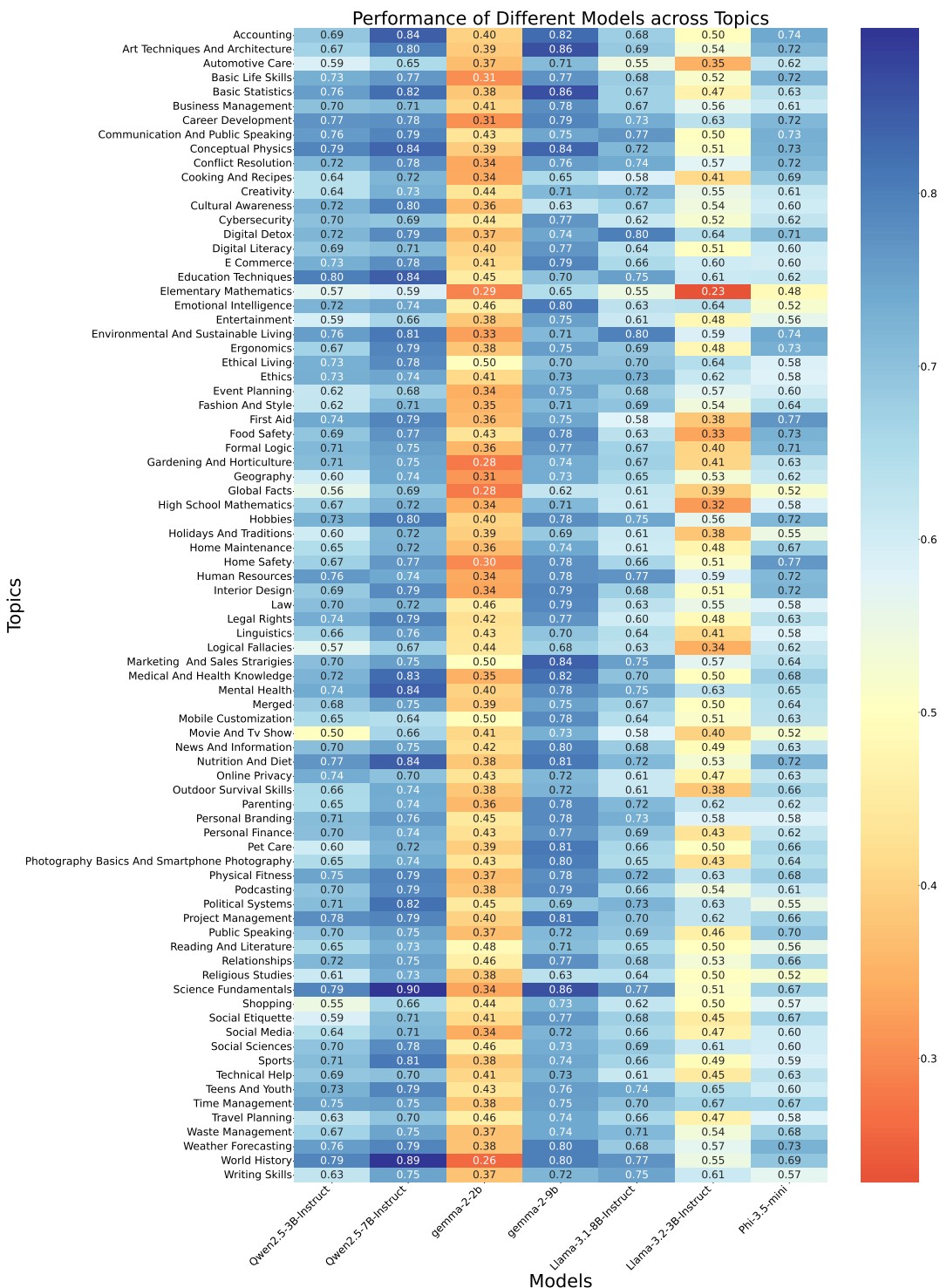

Figure 18: Heatmap visualizing the performance comparison of seven language models.

