# OpenReview forum: "Mobile-MMLU: A Mobile Intelligence Language Understanding Benchmark"
_DMLR — Accepted by DMLR_

### Review · Reviewer_WWy2 · 2025-10-01

**Recommendation:** 4
**Confidence:** 2

**Summary Of Contributions:**

The paper proposes Mobile-MMLU, a benchmark for evaluating LLMs in mobile-specific contexts. It introduces (i) Mobile-MMLU with 16,186 questions across 80 mobile-relevant domains, and (ii) Mobile-MMLU-Pro, a harder, multi-model-consistent subset of 9,497 questions. The dataset is constructed through a pipeline combining LLM generation, similarity filtering, human verification, and consensus-based ground truth refinement.

The authors conduct extensive experiments with small- to mid-scale LLMs (1B–9B) under both multiple-choice and open-ended formats, alongside on-device hardware tests (e.g., latency, memory, throughput on iPhone 14). Results show that Mobile-MMLU better discriminates between models than existing benchmarks (MMLU/MMLU-Pro), and that strong performance on traditional benchmarks does not always transfer to mobile contexts.

Overall, the work contributes a mobile-focused, order-invariant benchmark that jointly evaluates knowledge, reasoning, and efficiency, filling a critical gap in LLM evaluation for on-device AI.

**Strengths:**

1. Significance & Novelty: Introduces the first large-scale benchmark explicitly tailored for mobile-specific LLM evaluation, addressing a clear gap left by MMLU and related benchmarks.
2. Relation to Prior Work: Positions itself well with respect to MMLU, MMLU-Pro, and other mobile AI benchmarks (e.g., HammerBench, MobileAIBench), highlighting complementary rather than overlapping contributions.
3. Relevance: Highly relevant for the growing community working on on-device AI, edge intelligence, and LLM deployment in resource-constrained environments.
4. Research Quality: Dataset construction is rigorous, combining LLM generation, embedding-based similarity filtering, human verification, and multi-model consensus voting. This methodology reduces bias and ensures higher-quality ground truth.
5. Evaluation Depth: Goes beyond accuracy by incorporating open-ended evaluation and on-device hardware tests (latency, memory, throughput), providing a more holistic assessment of mobile feasibility.
6. Clarity: The paper is generally well-organized with clear motivation, detailed pipeline diagrams, and illustrative examples that make the benchmark construction transparent.
7. Ethical & Social Implications: Explicitly acknowledges limitations (e.g., time-sensitive queries, English-centric scope) and commits to dataset updates, reflecting responsible dataset stewardship.

**Audience:**

Yes

**Claims And Evidence:**

Overall, the paper’s main claims are well-supported by evidence. The evidence is convincing and presented with sufficient clarity. Minor limitations exist (e.g., reliance on LLM consensus for ground truth, hardware results on a single device), but these do not undermine the central claims.

**Datasets And Benchmarks:**

The dataset and benchmark are described with sufficient detail, are publicly hosted, and accompanied by ethical and maintenance considerations. While broader multilingual support and device diversity would strengthen future iterations, the current release meets DMLR’s requirements for dataset/benchmark submissions.

**Extended Submissions:**

N/A

**Limitations:**

1. Time-sensitive content: Around 3–4% of questions (e.g., app usage, population stats) may become outdated, requiring periodic dataset updates to maintain reliability.
2. Multiple-choice format bias: Despite efforts to mitigate length/order bias, reliance on MCQs may oversimplify real-world mobile interactions such as conversational, multi-turn, or context-dependent queries.
3. Limited multilingual scope: The benchmark is primarily English-based, which restricts applicability for non-English or culturally diverse mobile user populations.
4. Annotation reliance on LLM consensus: Ground truth refinement heavily depends on agreement among GPT-4o, Claude-3.5, and Gemini-2.0; human verification, though present, is partial.

**Requested Changes:**

1. Address time-sensitive questions:
Some queries (e.g., app-specific or population-related) risk becoming outdated. The authors should clarify the update schedule and consider mechanisms for automatic refresh or versioning of Mobile-MMLU.

2. Clarify annotation balance:
The paper relies heavily on LLM consensus. Please provide more detail on the proportion of human annotation vs. automated filtering, and whether higher human verification is planned in future releases.

**Strengths And Weaknesses:**

Strengths

1. Clear motivation: Addresses a well-identified gap in evaluating LLMs for mobile-specific contexts.
2. Novel benchmark: Large-scale dataset (16k+ questions, 80 domains) with practical, everyday tasks rather than academic knowledge.
3. Rigorous construction pipeline: Combines LLM generation, similarity filtering, human verification, and model-consensus refinement.
4.Comprehensive evaluation: Includes both MCQ and open-ended formats, plus real on-device hardware performance tests.
5. Discriminative results: Benchmark surfaces larger performance gaps and better highlights trade-offs in mobile deployment.

Weaknesses

1. Multiple-choice focus: While practical, MCQ format may oversimplify real mobile interactions (multi-turn, conversational tasks).
2. Limited multilingual coverage: Benchmark appears largely English-centric, which may restrict global applicability.
3. Dataset validation: Heavy reliance on LLM consensus; human annotation percentage could be higher for stronger guarantees.

---

### Review · Reviewer_cqex · 2025-10-17

**Recommendation:** 3
**Confidence:** 2

**Summary Of Contributions:**

The authors present Mobile-MMLU, a new benchmark specifically designed to evaluate LLMs in mobile contexts. They've created 16,186 multiple-choice questions across 80 topics they argue are particularly relevant to mobile users (things like travel planning, first aid, cooking recipes), plus a "Pro" version with 9,497 harder questions. They test various small models (1B-9B parameters) and include actual on-device performance metrics from iPhones, which is nice to see.

**Strengths:**

The paper makes a timely and relevant contribution by introducing a benchmark tailored to mobile LLMs. It offers broad coverage (80 categories, 16k+ questions) and a challenging Pro subset, with careful design choices to ensure rigor and consistency. Results across diverse models highlight its value, and the dataset’s open release supports reproducibility.

**Audience:**

Yes

**Broader Impact Concerns:**

The authors have included an "Ethical Statements and Limitations" section that sufficiently addresses the broader impact of the work.

**Claims And Evidence:**

The paper’s claims are well supported. The evidence (performance comparisons, order-bias experiments, hardware benchmarks) is thorough and convincing.

**Datasets And Benchmarks:**

The dataset is well described, and available on HuggingFace. The construction pipeline is transparent and reproducible.

**Extended Submissions:**

As far as I can tell, this is an original submission rather than an extension of previous work.

**Limitations:**

The authors discuss the limitations, but a few points stand out. The reliance on multiple-choice questions does not reflect the open-ended and multimodal nature of real mobile use. About 3.5% of the questions are time-sensitive and may quickly become outdated. The dataset is also English-only and shaped by a Western context, which limits its usefulness for global mobile users.

**Requested Changes:**

The following suggestions, while not critical, would help strengthen the paper:

1. The plan to “regularly update” the ~3.5% of time-sensitive questions is good in principle, but it feels a bit vague. It would help to have a short subsection on data maintenance, with a clear update schedule and maybe a way for the community to suggest changes.

2. The Open-Style Question (OSQ) evaluation is also a nice addition, but the paper doesn’t explain how the open-ended answers were graded; even a short description would make it clearer and easier to reproduce.

3. It would be useful for the authors to state explicitly in the main text that the benchmark is English-only, so readers have a clear sense of scope.

**Strengths And Weaknesses:**

Strengths:
There is clear motivation for mobile-centric evaluation of LLMs, and the authors provide a well-constructed dataset. The breadth of coverage (80 categories) and the methodological care taken (bias mitigation, order-invariance, human verification) make the benchmark robust. The addition of hardware-level evaluation is particularly valuable.

Weakness:
The dataset is mostly constrained to multiple-choice QA, which does not fully capture the richness of mobile interactions. Around 3–4% of questions are time-sensitive and may become outdated, raising long-term maintenance challenges. The dataset is also English-centric, leaving open the question of how well it generalizes to global mobile users. Finally, the broader ethical implications (e.g., personalization, privacy in on-device AI) are only briefly addressed.

---

### Review · Reviewer_mgHw · 2025-11-03

**Recommendation:** 3
**Confidence:** 2

**Summary Of Contributions:**

This paper introduces Mobile-MMLU, a new large-scale benchmark designed to evaluate the performance of LLMs specifically in mobile contexts. The authors argue that existing benchmarks like MMLU are tailored for desktop or server environments and fail to capture the unique constraints (e.g., limited resources) and user interaction patterns (e.g., practical, on-the-go tasks) of mobile devices.

**Strengths:**

See Strengths And Weaknesses

**Audience:**

Yes

**Broader Impact Concerns:**

The social impact and ethical considerations are worth more discussion. Some related works (such as [1]) may be helpful for the authors. It is recommended that the authors may provide more formal discussions with the guidance of these related works (such as [1]) if possible.

[1] Evaluating the Social Impact of Generative AI Systems in Systems and Society https://arxiv.org/pdf/2306.05949v2

**Claims And Evidence:**

See Strengths And Weaknesses

**Datasets And Benchmarks:**

See Strengths And Weaknesses

**Extended Submissions:**

n/a

**Limitations:**

See Strengths And Weaknesses

**Requested Changes:**

See Strengths And Weaknesses

**Strengths And Weaknesses:**

Pros:
1. The paper addresses a significant and growing gap in LLM evaluation. As AI shifts towards on-device deployment ("Edge AI"), having a benchmark that reflects mobile-specific challenges is crucial.
2. The data construction pipeline is well-designed and thoroughly explained. The use of multi-model consensus for filtering and establishing ground truth, combined with human verification, lends high credibility to the dataset's quality.
3. The authors go beyond simple accuracy metrics. By including on-device hardware performance (inference speed, memory usage) and evaluating under a stricter OSQ format, they provide a more comprehensive picture of a model's suitability for mobile deployment.

Cons:
1. The on-device performance tests were conducted exclusively on an iPhone 14, a high-end device. The mobile ecosystem is incredibly diverse, with a vast range of Android devices featuring different SoCs (e.g., Snapdragon, MediaTek), RAM configurations, and NPUs. Performance on a flagship Apple device may not be representative of performance on a mid-range or budget Android phone, which constitutes a large portion of the market. Could you discuss the choice of the iPhone 14 for hardware tests? Are there plans to expand the hardware evaluation to include a diverse set of Android devices to provide a more complete picture of on-device performance?
2. The process for creating Mobile-MMLU-Pro filters out questions that small models answer correctly. One could argue that a key measure of a good mobile model is its ability to reliably and efficiently handle common, often "easy," user queries. Filtering these out might skew the evaluation away from practical utility towards more complex, less frequent tasks. Could you elaborate on the trade-offs of removing "easy" questions for the Pro version? Does this risk undervaluing models that are highly optimized for quickly and accurately handling the most common user tasks?
3. For the 5.8% of questions with multiple correct options, the ground truth was determined by a voting process among large models. How do you mitigate the risk of these models reaching a consensus that is factually incorrect or suboptimal, especially for questions whose answers may change over time (e.g., app functionalities)?
4. Given the limitations of the MCQ format, do you envision future versions of Mobile-MMLU incorporating more interactive or conversational tasks to better simulate real-world mobile assistant usage?